# PatchMoE: A Time Series Foundation Model with Hierarchical Patch-wise Mixture-of-Experts

## Abstract

Recently, time series foundation models (TSFMs) pre-trained on massive datasets have achieved remarkable zero-shot performance. However, effectively modeling the diverse *inter*-series and *intra*-series patterns in large-scale datasets remains a significant challenge. Most existing methods, constrained by a single, fixed tokenizer, lack the flexibility to capture the pattern diversity. To tackle this issue, we introduce PatchMoE, a novel hierarchical Mixture of Experts (MoE) architecture, comprising Patch-wise Experts and Sample-wise Hierarchical Router as key components. Specifically, Patch-wise Experts are employed to capture diverse *inter*-series patterns with specialized patch tokenizers. While Sample-wise Hierarchical Router tackles *intra*-series patterns by dispatching the entire sample to experts. This process allows each sample to undergo hierarchical routing through multiple MoE layers, where each layer gradually outputs a partial forecast. Furthermore, to address the efficiency bottleneck of MoE architecture, we develop a highly efficient training framework for the time series modality based on Megatron-LM[1], which implements expert parallelism and achieves a $3\times$ to $5\times$ training speedup under identical experimental settings. Benefiting from this, for the first time, we scale a time series foundation model to 8.5 billion parameters, achieving state-of-the-art results on zero-shot forecasting tasks. Compared with dense and sparse models of equivalent scale of parameters, PatchMoE demonstrates significant improvements in both effectiveness and efficiency.

## 1 Introduction

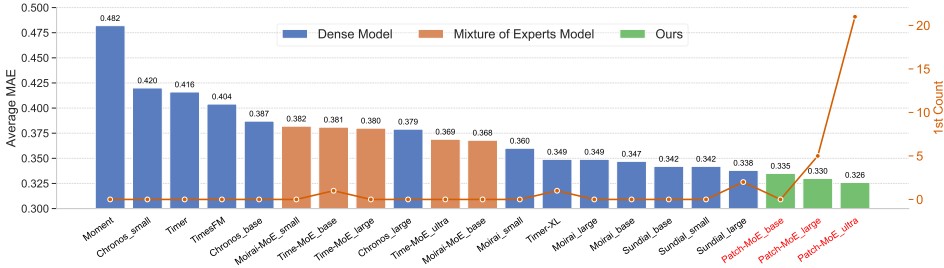

Figure 1: PatchMoE achieves state-of-the-art zero-shot performance on the long-term forecasting (LTF) benchmark Wu et al. (2021)[2].

Time series forecasting is a critical task for a multitude of real-world applications across diverse domains, including energy, weather, retail sales, etc Li et al. (2025). With the recent proliferation of massive time series data, there has been a surge of interest toward developing Time Series Foundation Models (TSFMs). These models are pre-trained on time series corpora containing billions or even trillions of time points Liu et al. (2025e); Xiaoming et al. (2025), enabling them to provide *Zero-Shot* forecasting capabilities across diverse domains.

---

[1] https://github.com/NVIDIA/Megatron-LM
[2] Zero-shot results except for Moirai-MoE are obtained from Liu et al. (2025e); Xiaoming et al. (2025); Liu et al. (2025d).

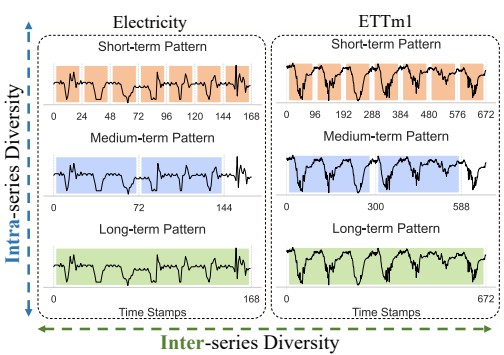 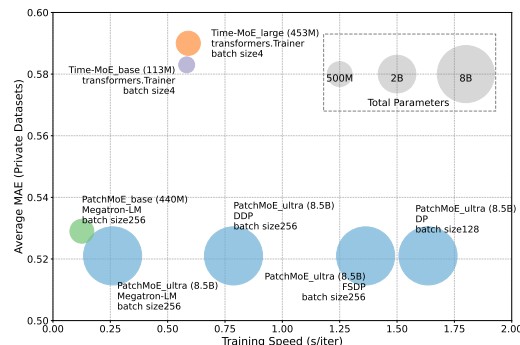

Figure 2: Illustration of Inter-series and Intra-series pattern diversity in time series data.

Figure 3: Model efficiency and parameters comparison between PatchMoE and Time-MoE[1].

A central component in TSFMs is the *patch tokenizer*, an instrumental mechanism first introduced by PatchTST Nie et al. (2023), which enriches semantics of each token and accelerates model training by compressing the input sequence length. However, most existing TSFMs employ a single tokenizer with fixed patch length Goswami et al. (2024); Das et al. (2024); Liu et al. (2025d;e). This one-size-fits-all approach struggles to handle the diverse patterns existing both *inter-* and *intra-* series, especially when models are trained on large-scale data from different domains Chen et al. (2024); Zhang et al. (2024a;b). As illustrated in Figure 2, *inter*-series diversity is evident, as different series from Electricity[2] and ETTm1 Zhou et al. (2021) exhibit fundamentally distinct patterns with significantly different temporal spans. Additionally, *intra*-series diversity arises from the co-existence of diverse patterns within a single series, presenting vastly different characteristics under varying time scales. Both challenges demonstrate that no single patch length can be universally optimal. This strongly motivates the need for adaptive tokenization—a mechanism that can dynamically adjust its scale to match the specific *inter-* and *intra*-series patterns present in the data.

The Mixture-of-Experts (MoE) Jiang et al. (2024) architecture provides an alternative method for adaptive tokenization, reframing the selection of optimal tokenizers as an expert-routing problem. In contrast to dense models, sparsely activated MoE models enable substantial model scaling and deliver competitive performance Dai et al. (2024). However, applying the MoE architecture to time series modality is a non-trivial task. Pioneering works, such as Time-MoE Xiaoming et al. (2025) and Moirai-MoE Liu et al. (2025b) extend the MoE architecture to the time series domain, typically employing a modality-agnostic design where experts are standard Feed-Forward Networks (FFNs). A fundamental limitation of this approach is that it fails to explicitly model the unique characteristics of time series as mentioned above. Consequently, it remains a critical question how to design an MoE architecture that can effectively handle the complex patterns of time series.

To this end, we introduce PatchMoE, a novel hierarchical Mixture of Experts (MoE) architecture, comprising two core components: **Patch-wise Experts** and **Sample-wise Hierarchical Router**. Specifically, Patch-wise Experts are employed to capture diverse *inter*-series patterns. Each expert is a stack of multiple Transformer Vaswani et al. (2017) layers equipped with a specialized patch tokenizer to model correlations at a particular temporal scale. Concurrently, Sample-wise Hierarchical Router tackles *intra*-series diversity. Instead of routing tokens adopted in conventional MoE, our router dispatches the entire sample to experts. Each sample undergoes sequential routing through multiple MoE layers. Each MoE layer outputs a partial forecast, and the final prediction is aggregated from the forecasts of all MoE layers. Together, these two mechanisms enhance the capability for PatchMoE to effectively handle large-scale time series data.

Despite the advantages of MoE, its scalability in time series domain has been crippled by training inefficiency, largely due to the serialization of expert computations in existing frameworks (e.g., Time-MoE, Moirai-MoE). This approach creates a severe throughput bottleneck and prevents effective scaling. To address this challenge, we develop an efficient pre-training framework adapted for the time series domain based on Megatron-LM Shoeybi et al. (2019), which implements *expert*

---

[1]For Time-MoE, increasing the micro-batch size beyond 4 results in an out-of-memory (OOM) error.

[2]https://archive.ics.uci.edu/ml/datasets/ElectricityLoadDiagrams20112014

*parallelism* and supports flexible weighted sampling across large-scale time series corpus with over 300B time points. As shown in Figure 3, our framework achieves about $3\times$ to $5\times$ training speedup over other frameworks. When comparing with models of a similar scale, PatchMoE$_{base}$(440M) not only trains approximately $3\times$ faster, but also achieves a substantially better performance than Time-MoE$_{large}$(453M). This breakthrough in efficiency allows us, for the first time, to scale the TSFM to $8.5$ billion parameters.

Extensive experiments demonstrate the effectiveness of PatchMoE. Our model achieves state-of-the-art zero-shot results on well-acknowledged long-term forecasting (LTF) benchmarks Wu et al. (2021), as shown in Figure 1. Furthermore, PatchMoE shows dominant zero-shot performance on three commercially valuable real-world datasets, drawn from the e-commerce and travel domains. To foster future research, we open-source these datasets, providing the community with a broader benchmark for zero-shot evaluation. Our contributions can be summarized as follows:

- We propose PatchMoE, a novel hierarchical MoE architecture for time series modality, which comprises Patch-wise Experts and Sample-wise Hierarchical Router as key components to effectively handle diverse *inter*-series and *intra*-series patterns in large-scale datasets.

- We develop an efficient pre-training framework based on Megatron-LM. By implementing expert parallelism, our framework boosts $3\times$ to $5\times$ training speed, and supports flexible weighted sampling across large-scale time series corpus with over 300B time points. This breakthrough in efficiency allows us, for the first time, to scale the TSFM to $8.5$ billion parameters.

- PatchMoE achieves state-of-the-art zero-shot performance on well-acknowledged long-term forecasting (LTF) benchmarks. Furthermore, we introduce three novel real-world datasets from the high-value commercial domains for future research. Our model also demonstrates dominant performance on these new datasets.

## 2 RELATED WORK

**Time Series Foundation Models.** TSFMs aim to achieve strong zero-shot generalization by *natively* pre-training on massive data Li et al. (2025). While initial efforts focused on adapting either auto-regressive decoder-only architectures Das et al. (2024); Liu et al. (2024b) or bidirectional encoder-only models for masked reconstruction Goswami et al. (2024); Woo et al. (2024), both approaches face computational bottlenecks at scale. Addressing this challenge, Time-MoE Xiaoming et al. (2025) and Moiral-MoE Liu et al. (2025b) have emerged as pioneering frameworks, which integrate a sparse Mixture-of-Experts (MoE) architecture at their core, and achieve a breakthrough in efficiency and capability. These MoE-based advancements, alongside progress in highly efficient lightweight models Wang et al. (2025b) and probabilistic generative frameworks Liu et al. (2025e), represent the current architectural frontier for TSFMs. Building on these advancements, we further explore how to combine the unique characteristics of time series to scale up to larger TSFMs.

**Patch-based Time Series Forecasting.** Patch-based methods Nie et al. (2023); Liu et al. (2024b); Das et al. (2024) are a cornerstone of modern time series forecasting, but to handle diverse *inter*-series and *intra*-series patterns in time series data, strategies have evolved from fixed-size patches to complex multi-scale and adaptive designs, such as multi-scale modeling via parallel branches Zhang et al. (2024b) or hierarchical decomposition Zhong et al. (2024); Ekambaram et al. (2024); data-driven adaptive strategies based on intrinsic periods Woo et al. (2024); Tang & Zhang (2025); Wang et al. (2025b) or dynamic routing Chen et al. (2024); and even advanced methods for fully learned, non-uniform segmentation via self-supervised learning Prabhakar Kamarthi & Prakash (2024). To address the trade-off between fixed-size patching and flexible yet computationally complex patching strategies, and to achieve a balance between computational efficiency and model capacity, we employ an MoE architecture, which integrates Patch-wise Experts to enable adaptive patching.

**Mixture of Experts for Time Series Forecasting.** The MoE architecture has proven to be highly effective for scaling TSFMs, offering a compelling solution to handle large-scale time series data Xiaoming et al. (2025); Liu et al. (2025b). A distinct line of work has explored alternative routing strategies and expert designs beyond the token-level MoE. For instance, MoLE Ni et al. (2024) implements sequence-level routing by using the initial timestamp to determine the weights for combining several linear experts. Other works, such as FreqMoE Liu (2025) and MoFE-Time Liu et al. (2025c), have focused on designing experts that specialize in data properties rather than abstract

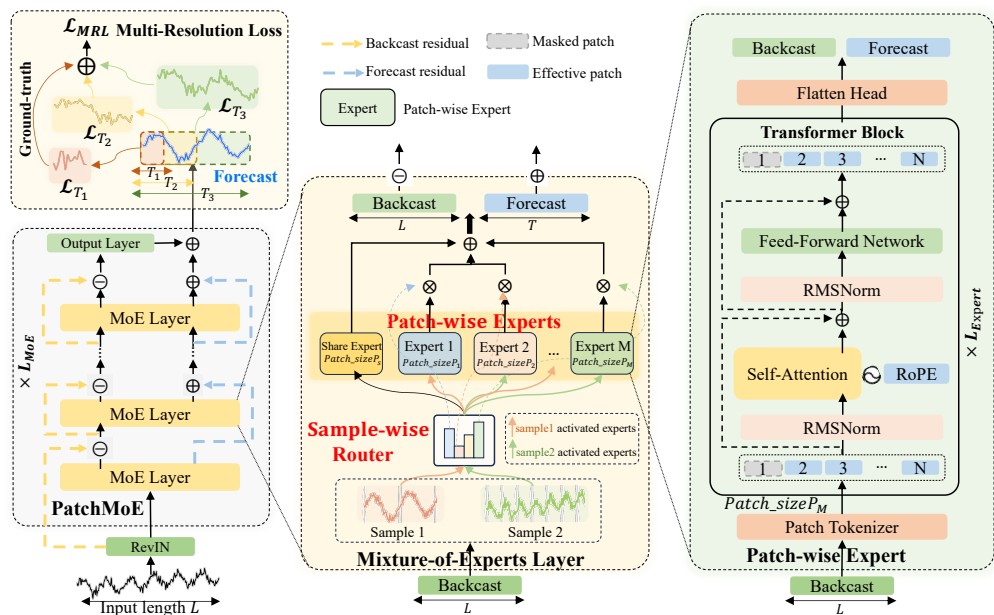

Figure 4: The overview of PatchMoE, a hierarchical MoE architecture with $L_{\mathrm{MoE}}$ layers stacked via backcast residual connections ($\ominus$) and forecast aggregation connections ($\oplus$). The core components of the MoE layer consist of: 1) Sample-wise Router that hierarchically dispatches backcast sample to experts; 2) Patch-wise Experts with each of them comprising a specialized patch tokenizer and $L_{\mathrm{Expert}}$ Transformer layers. PatchMoE is optimized by minimizing the Multi-Resolution Pre-training Loss, which enhances proficiency across different forecast horizons during inference.

patterns. These approaches highlight a critical design choice in MoE: the nature of expert specialization. Therefore, our work diverges by proposing that experts should specialize in processing time series at different structural scales, which we achieve through Sample-wise Hierarchical Router.

## 3 METHOD

Figure 4 illustrates the overall structure of PatchMoE. In this section, we first formulate the problem in Section 3.1. We then detail the methodology of PatchMoE in Section 3.2, and finally present our pre-training framework in Section 3.3.

### 3.1 PROBLEM DEFINITION.

Let $L$ and $T$ be the maximum context length and the forecast horizon of the model, respectively. Each input sample is a univariate series $\boldsymbol{x} = \{x_1, x_2, ..., x_L\} \in \mathbb{R}^L$, and $\boldsymbol{y} = \{y_1, y_2, ..., y_T\} \in \mathbb{R}^T$ is the corresponding target spanning forecast horizon of $T$ time points. The objective is to train a unified model, $\mathcal{F}$, which takes the input $\boldsymbol{x}$ to produce a prediction $\boldsymbol{y}' = \mathcal{F}(\boldsymbol{x})$, such that a chosen loss function $\mathcal{L}(\boldsymbol{y}', \boldsymbol{y})$ is minimized.

### 3.2 PATCHMOE

As illustrated in Figure 4, PatchMoE leverages a residual stacking framework with MoE layers to dynamically capture multi-scale temporal dependencies. The input series $\boldsymbol{x} \in \mathbb{R}^L$ is first processed by Reversible Instance Normalization (RevIN) Kim et al. (2021) to mitigate distribution shifts. The normalized series $\boldsymbol{x}'$ is then fed into a stack of $L_{\mathrm{MoE}}$ identical MoE layers, which consists of a Sample-wise Router and Patch-wise Experts.

**Hierarchical Modeling**. The stack of MoE layers employs a backcast-forecast *doubly residual stacking* principle Oreshkin et al. (2020); Challu et al. (2023). Let $\boldsymbol{b}^{(l-1)} \in \mathbb{R}^L$ and $\boldsymbol{f}^{(l-1)} \in \mathbb{R}^T$

be the backcast and forecast inputs to the $l$-th MoE layer ($l \in [1, L_{\text{MoE}}]$), respectively, with initial inputs set as $\boldsymbol{b}^{(0)} = \boldsymbol{x}'$ and $\boldsymbol{f}^{(0)} = \boldsymbol{0}$ when $l = 1$. The $l$-th MoE layer, denoted as $\text{MoELayer}^{(l)}$, processes $\boldsymbol{b}^{(l-1)}$ to produce a partial backcast $\Delta \boldsymbol{b}^{(l)}$ and a partial forecast $\Delta \boldsymbol{f}^{(l)}$:

$$\Delta \boldsymbol{b}^{(l)}, \Delta \boldsymbol{f}^{(l)} = \text{MoELayer}^{(l)}(\boldsymbol{b}^{(l-1)}). \tag{1}$$

The backcast $\boldsymbol{b}^{(l)} \in \mathbb{R}^L$, which serves as the input to the next MoE layer, is computed via a residual connection. This process be deemed as running a sequential analysis of the input signal. While the final output of the model $\boldsymbol{y}' \in \mathbb{R}^T$ is obtained by aggregating the forecasts from all MoE layers:

$$\boldsymbol{b}^{(l)} = \boldsymbol{b}^{(l-1)} - \Delta \boldsymbol{b}^{(l)}, \quad \boldsymbol{y}' = \sum_{l=1}^{L_{\text{MoE}}} \Delta \boldsymbol{f}^{(l)}. \tag{2}$$

**Sample-wise Router.** To enable adaptive tokenization, we employ a sample-wise router that selects a *sparse* combination of experts for each input instance. Following the DeepSeekMoE paradigm Dai et al. (2024), the router directs path to $N$ experts, in addition to a *Shared Expert* with a shared patch size $P_s$ which is active for all samples and provides a robust baseline representation. For a given backcast $\boldsymbol{b}^{(l-1)} \in \mathbb{R}^L$, the router computes scores $\boldsymbol{s} \in \mathbb{R}^N$ via a linear projection: $\boldsymbol{s} = \boldsymbol{W}_r \boldsymbol{b}^{(l-1)}$, where $\boldsymbol{W}_r \in \mathbb{R}^{N \times L}$. Then a Top-$k$ gating mechanism selects $k$ experts with the highest scores. The final outputs of the MoE layer are computed by adding the weighted sum outputs of the chosen experts and the output of the Shared Expert:

$$\Delta \boldsymbol{b}^{(l)} = \Delta \boldsymbol{b}_{\text{shared}}^{(l)} + \sum_{j=1}^{N} g_j \cdot \Delta \boldsymbol{b}_j^{(l)}, \quad \Delta \boldsymbol{f}^{(l)} = \Delta \boldsymbol{f}_{\text{shared}}^{(l)} + \sum_{j=1}^{N} g_j \cdot \Delta \boldsymbol{f}_j^{(l)}, \tag{3}$$

where $g_j$ is the normalized gating weight for the active $j$-th expert and zero otherwise.

To alleviate the load-imbalance problem caused by MoE training, which can lead to router collapse, we follow the approaches of Xiaoming et al. (2025); Dai et al. (2024) to achieve load balancing at the sample level by introducing an auxiliary loss:

$$\mathcal{L}_{\text{aux}} = N \sum_{j=1}^{N} f_j r_j, \quad f_j = \frac{1}{kB} \sum_{i=1}^{B} \mathbb{I}(\text{Sample } i \text{ selects Expert } j), \quad r_j = \frac{1}{B} \sum_{i=1}^{B} s_{i,j}, \tag{4}$$

where $\mathbb{I}$ is the indicator function, $B$ denotes the global batch size, $f_j$ and $r_j$ represent the fraction of samples and the proportion of router probability allocated to the $j$-th expert, respectively.

**Patch-wise Experts.** As detailed in the right panel of Figure 4, each expert is a stack of $L_{\text{Expert}}$ identical Transformer layers Vaswani et al. (2017), and is distinguished by a specialized tokenizer with a unique patch size. Each Transformer layer, built upon a Pre-LN architecture Touvron et al. (2023), comprises a non-causal Multi-Head Self-Attention (MHSA) block and a Feed-Forward Network (FFN) with a SwiGLU activation Shazeer (2020). The layer incorporates RMSNorm Zhang & Sennrich (2019) for normalization and RoPE Su et al. (2024) for positional information, with all bias terms omitted. After stacking $L_{\text{Expert}}$ layers, a flatten head produces the final output of the expert. Details description of each module can be found in Appendix A.1.

**Multi-Resolution Loss.** To ensure proficiency across arbitrary forecast horizons, we employ a Multi-Resolution Loss, as detailed in the top-left part of Figure 4. Specifically, the forecast horizon $T$ is partitioned into $H$ sub-horizons $\{T_1, T_2 \dots, T_H\}$, where $T_1 < T_2 < \cdots < T_H$. We compute a separate Mean Absolute Error (MAE) loss $\mathcal{L}_{T_h}$ for each sub-horizon $T_h$, where $h \in [1, H]$. The multi-resolution loss $\mathcal{L}_{\text{MRL}}$ is the sum of these resolution-specific losses:

$$\mathcal{L}_{\text{MRL}} = \sum_{h=1}^{H} \mathcal{L}_{T_h} = \sum_{h=1}^{H} \frac{1}{|T_h|} |\hat{\boldsymbol{y}}_{T_h} - \boldsymbol{y}_{T_h}|, \tag{5}$$

where $\boldsymbol{y}_{T_h}$ and $\hat{\boldsymbol{y}}_{T_h}$ denote the first $P_h$ time points of the target and the output, respectively. This objective enables the model to achieve balanced accuracy across various prediction horizons, resulting in robust and versatile predictions. Finally, we combine $\mathcal{L}_{\text{aux}}$ with $\mathcal{L}_{\text{MRL}}$ via a coefficient $\alpha$. The overall loss function for pre-training PatchMoE is thus defined as: $\mathcal{L} = \mathcal{L}_{\text{MRL}} + \alpha \mathcal{L}_{\text{aux}}$.

## 3.3 Pre-training Framework

We present the key designs of our pre-training pipeline for PatchMoE, which features: 1) a domain diversity sampling strategy; 2) a masking mechanism to enable arbitrary input length; 3) expert parallelism to boost training efficiency.

**Domain Diversity Sampling.** Leveraging the sampling process of the Megatron-LM framework, we extend it to time series modality, enabling efficient and custom domain-weighted sampling on massive corpora. According to pre-defined weights, we mitigate domain imbalance in the time series data and ensure sample diversity. The sampling process is detailed in Appendix A.2.

**Input Mask.** To enable our model to process time series of variable input lengths, we employ a random left-padding strategy after our sampling process. Specifically, for each input sample of length $L$, we first sample a random integer $M$ from the uniform distribution over $[0, L)$. We then pad the first $M$ time points of the sample with a predefined padding value. Samples already padded to length $L$ during the sampling process (see Appendix A.2) are exempted from this step. These padded positions are subsequently masked out in the attention mechanism, thereby ensuring that the model can handle variable-length inputs without being influenced by the padded values.

**Expert Parallelism.** The scalability of MoE is often hindered by the training inefficiency of *serialized expert computations*. To overcome this bottleneck, we implement an expert parallelism strategy for PatchMoE, building upon the Megatron-LM framework. The set of GPUs that collectively hosts the complete set of experts for an MoE layer constitutes an expert parallel group, with each GPU holding equivalent number of distinct experts. The forward pass employs a two-stage *all-to-all* communication pattern. After the router assigns samples to experts, an *all-to-all* communication first dispatches the samples to the GPUs hosting their designated experts. Following *parallel computation* across all experts, a second *all-to-all* communication gathers the outputs back to their source GPUs. This dispatch-compute-gather cycle is repeated for each MoE layer, enabling massive parallelism and significantly improving training throughput.

## 4 EXPERIMENTS

### 4.1 IMPLEMENTATION DETAILS

Prior to our work, the largest existing TSFM is Time-MoE Xiaoming et al. (2025). To ensure a fair comparison, we introduce PatchMoE$_{\text{large}}$ (1.1B/2.5B), and PatchMoE$_{\text{base}}$ (200M/453M), to closely align with Time-MoE$_{\text{ultra}}$ (1.1B/2.4B) and Time-MoE$_{\text{large}}$ (200M/453M), respectively. Furthermore, we scale PatchMoE up to 8.5B with 3.8B average activated parameters to obtain PatchMoE$_{\text{ultra}}$, which stands as the largest time series foundation model to date. The detailed configurations of these models are summarized in Table 1. Leveraging our efficient training framework, all models are pre-trained for 50,000 iterations with a global batch size of 4096. The expert parallelism size is set to 4. For our Multi-Resolution Loss, the sub-horizons are set to $\{24, 96, 336\}$. Additional implementation details are provided in the Appendix B.

### 4.2 MAIN RESULTS

PatchMoE consistently outperforms baseline models by large margins on LTF benchmarks and our proposed datasets. To ensure a fair comparison, we adhered to the configurations of Xiaoming et al. (2025) for the zero-shot and full-shot forecasting with a unified evaluation pipeline. Specifically, we evaluate zero-shot performance against 7 TSFMs, categorized as MoE-based models and dense models. Furthermore, for the full-shot evaluation, we benchmark PatchMoE against 9 models from three distinct families: Classical deep learning models, MoE-based models, and multi-patches based models. Description of all benchmarks and baselines can be found in Appendix C.

**Zero-shot Forecasting.** As shown in Table 2, PatchMoE$_{\text{ultra}}$ scaled up to 3.8B/8.5B parameters delivers state-of-the-art zero-shot results, with the best performance in 58 cases out of the overall 90 cases, surpassing both leading dense and other MoE-based models. When compared with MoE-based models, PatchMoE surpass Time-MoE with closely equivalent parameter scales, with PatchMoE$_{\text{large}}$ (1.2B/2.5B, 0.330) outperforming Time-MoE$_{\text{ultra}}$ (1.1B/2.4B, 0.369), and

Table 1: Model Configurations of PatchMoE.

| Model | $L_{\text{MoE}}$ | $L_{\text{Expert}}$ | Heads | $d_{\text{model}}$ | $d_{\text{ff}}$ | $L$ | Experts | $k$ | Avg. Activated Params | Total Params |
|---|---|---|---|---|---|---|---|---|---|---|
| PatchMoE$_{\text{base}}$ | 2 | 4 | 8 | 512 | 2048 | 1440 | 4 | 1 | 200M | 440M |
| PatchMoE$_{\text{large}}$ | 2 | 4 | 16 | 1024 | 4096 | 2880 | 4 | 1 | 1.2B | 2.5B |
| PatchMoE$_{\text{ultra}}$ | 3 | 4 | 16 | 1024 | 4096 | 2880 | 8 | 2 | 3.8B | 8.5B |

Table 2: Zero-shot performance of TSFMs on LTF benchmarks and our proposed datasets. Averaged results across four prediction horizons $\{96, 192, 336, 720\}$ are reported here. A lower MSE or MAE is better. $1^{st}$ Cnt denotes the total number of times a model ranks first across all prediction lengths and datasets. The subscripts $b$, $l$, and $u$ denote base, large, and ultra, respectively. Results of datasets included in pre-training procedure are denoted by a dash$(-)$. Results of unreleased models that cannot be evaluated on our proposed datasets are denoted by a slash$(\backslash)$. **Red**: the best, Blue: the 2nd best. Detailed results are included in Table 6.

| Models | Ours | | | | | | MoE-based models | | | | | | | | | | Dense baselines | | | | | | | |
|---|---|---|---|---|---|---|---|---|---|---|---|---|---|---|---|---|---|---|---|---|---|---|---|---|
| | PatchMoE$_b$ | | PatchMoE$_l$ | | PatchMoE$_u$ | | Time-MoE$_l$ | | Time-MoE$_u$ | | Moirai-MoE$_b$ | | Sundial$_b$ | | Sundial$_l$ | | Timer-XL | | Moirai$_l$ | | Chronos$_l$ | | TimesFM | |
| Metrics | MSE | MAE | MSE | MAE | MSE | MAE | MSE | MAE | MSE | MAE | MSE | MAE | MSE | MAE | MSE | MAE | MSE | MAE | MSE | MAE | MSE | MAE | MSE | MAE |
| ETTh1 | 0.414 | 0.416 | 0.430 | 0.418 | 0.420 | **0.413** | 0.394 | 0.420 | 0.404 | 0.421 | 0.507 | 0.436 | 0.411 | 0.434 | 0.395 | 0.420 | 0.404 | 0.417 | 0.480 | 0.440 | 0.588 | 0.466 | 0.473 | 0.444 |
| ETTh2 | 0.344 | 0.376 | 0.340 | 0.375 | 0.346 | **0.372** | 0.405 | 0.415 | 0.371 | 0.399 | 0.385 | 0.397 | **0.333** | 0.387 | 0.334 | 0.387 | 0.347 | 0.388 | 0.368 | 0.377 | 0.455 | 0.427 | 0.392 | 0.406 |
| ETTm1 | 0.377 | 0.380 | 0.347 | 0.372 | 0.332 | **0.366** | 0.376 | 0.406 | 0.356 | 0.392 | 0.461 | 0.464 | 0.336 | 0.377 | **0.331** | 0.369 | 0.373 | 0.392 | 0.422 | 0.391 | 0.556 | 0.465 | 0.433 | 0.419 |
| ETTm2 | 0.280 | 0.326 | 0.258 | **0.309** | 0.258 | 0.310 | 0.316 | 0.361 | 0.288 | 0.344 | 0.338 | 0.352 | 0.258 | 0.320 | 0.254 | 0.315 | 0.273 | 0.336 | 0.330 | 0.344 | 0.295 | 0.338 | 0.328 | 0.347 |
| Weather | 0.241 | 0.264 | 0.236 | 0.260 | 0.230 | **0.257** | 0.270 | 0.300 | 0.256 | 0.288 | 0.287 | 0.292 | 0.234 | 0.270 | 0.238 | 0.275 | 0.256 | 0.294 | 0.264 | 0.273 | 0.279 | 0.306 | - | - |
| ECL | 0.161 | 0.250 | 0.156 | 0.242 | 0.150 | 0.237 | - | - | - | - | 0.188 | 0.266 | 0.169 | 0.265 | 0.166 | 0.262 | 0.174 | 0.268 | 0.186 | 0.270 | 0.204 | 0.273 | - | - |
| $1^{st}$ Cnt | 0 | | 5 | | **35** | | 1 | | 2 | | 0 | | 4 | | 15 | | 2 | | 0 | | 0 | | 0 | |
| Travel1 | 0.929 | 0.660 | 0.921 | 0.657 | 0.905 | 0.652 | 1.107 | 0.708 | \ | \ | 0.934 | 0.676 | 0.938 | 0.667 | \ | \ | 0.995 | 0.716 | 1.983 | 1.082 | 1.733 | 0.881 | 0.908 | 0.663 |
| Travel2 | 0.275 | 0.272 | 0.276 | 0.275 | 0.270 | 0.274 | 0.283 | 0.301 | \ | \ | 0.307 | 0.286 | 0.301 | 0.288 | \ | \ | 0.392 | 0.338 | 1.120 | 0.751 | 0.310 | 0.287 | 0.285 | 0.285 |
| E-comm | 2.548 | 0.654 | 2.466 | 0.648 | 2.435 | 0.637 | 3.037 | 0.761 | \ | \ | 2.902 | 0.698 | 2.553 | 0.690 | \ | \ | 2.667 | 0.751 | 4.620 | 1.381 | 2.565 | 0.697 | 2.665 | 0.686 |
| $1^{st}$ Cnt | 4 | | 0 | | **23** | | 2 | | \ | | 0 | | 0 | | \ | | 0 | | 0 | | 0 | | 1 | |

Table 3: Full-shot performance of PatchMoE and domain models on LTF benchmarks and our proposed datasets. Averaged results across four prediction horizons $\{96, 192, 336, 720\}$ are reported here. A lower MSE or MAE indicates a better prediction. **Red**: the best, Blue: the 2nd best. Detailed results are included in Table 7.

| Models | Ours | | | | | | Classic Models | | | | | | | | | | Multi-patches Models | | | | MoE-based Models | | | |
|---|---|---|---|---|---|---|---|---|---|---|---|---|---|---|---|---|---|---|---|---|---|---|---|---|
| | PatchMoE$_b$ | | PatchMoE$_l$ | | PatchMoE$_u$ | | PatchTST | | iTransformer | | TiDE | | DLinear | | TimeMixer | | Pathformer | | MTST | | MoLE | | FreqMoE | |
| Metrics | MSE | MAE | MSE | MAE | MSE | MAE | MSE | MAE | MSE | MAE | MSE | MAE | MSE | MAE | MSE | MAE | MSE | MAE | MSE | MAE | MSE | MAE | MSE | MAE |
| ETTh1 | 0.408 | 0.412 | 0.395 | 0.413 | **0.393** | **0.402** | 0.413 | 0.434 | 0.454 | 0.447 | 0.419 | 0.430 | 0.422 | 0.437 | 0.448 | 0.442 | 0.541 | 0.507 | 0.430 | 0.429 | 0.442 | 0.443 | 0.440 | 0.429 |
| ETTh2 | 0.336 | 0.367 | 0.337 | 0.371 | **0.325** | **0.366** | 0.331 | 0.381 | 0.383 | 0.407 | 0.345 | 0.394 | 0.431 | 0.447 | 0.365 | 0.395 | 0.344 | 0.379 | 0.348 | 0.385 | 0.377 | 0.414 | 0.367 | 0.396 |
| ETTm1 | 0.345 | 0.366 | 0.331 | **0.363** | 0.327 | 0.366 | 0.353 | 0.382 | 0.407 | 0.410 | 0.355 | 0.378 | 0.357 | 0.379 | 0.381 | 0.396 | 0.382 | 0.386 | 0.383 | 0.398 | 0.357 | 0.379 | 0.375 | 0.396 |
| ETTm2 | **0.247** | 0.304 | 0.249 | 0.304 | 0.247 | **0.301** | 0.256 | 0.317 | 0.288 | 0.332 | 0.249 | 0.312 | 0.267 | 0.332 | 0.275 | 0.323 | 0.273 | 0.316 | 0.279 | 0.323 | 0.263 | 0.318 | 0.271 | 0.338 |
| Weather | 0.229 | 0.260 | 0.224 | 0.255 | 0.219 | 0.254 | 0.226 | 0.264 | 0.257 | 0.278 | 0.226 | 0.264 | 0.246 | 0.300 | 0.240 | 0.271 | 0.239 | 0.263 | 0.255 | 0.278 | 0.236 | 0.273 | 0.248 | 0.276 |
| ECL | 0.156 | 0.246 | 0.153 | 0.241 | 0.152 | 0.240 | 0.159 | 0.253 | 0.178 | 0.270 | 0.159 | 0.252 | 0.166 | 0.264 | 0.182 | 0.275 | 0.182 | 0.269 | 0.187 | 0.277 | 0.172 | 0.266 | 0.179 | 0.271 |
| $1^{st}$ Cnt | 4 | | 12 | | **50** | | 2 | | 0 | | 1 | | 0 | | 0 | | 1 | | 0 | | 0 | | 0 | |
| Travel1 | 0.929 | 0.660 | 0.921 | 0.657 | **0.905** | **0.652** | 1.467 | 0.869 | 1.673 | 0.950 | 1.822 | 1.019 | 1.270 | 0.800 | 3.382 | 1.401 | 1.020 | 0.717 | 1.211 | 0.802 | 1.059 | 0.734 | 1.722 | 0.976 |
| Travel2 | 0.266 | 0.270 | 0.255 | **0.264** | 0.254 | 0.267 | 0.308 | 0.321 | 0.325 | 0.329 | 0.392 | 0.375 | 0.293 | 0.303 | 0.386 | 0.371 | 0.273 | 0.272 | 0.288 | 0.291 | 0.284 | 0.290 | 0.589 | 0.508 |
| E-comm | 2.412 | **0.615** | 2.368 | 0.619 | 2.376 | 0.631 | 2.530 | 0.740 | 2.786 | 0.815 | 3.238 | 0.932 | 2.501 | 0.740 | 3.940 | 1.029 | 2.557 | 0.687 | 2.484 | 0.705 | 2.522 | 0.716 | 3.854 | 1.198 |
| $1^{st}$ Cnt | 4 | | 11 | | **18** | | 0 | | 0 | | 0 | | 0 | | 0 | | 0 | | 0 | | 0 | | 0 | |

PatchMoE$_{base}$ (200M/440M, 0.335) outperforming Time-MoE$_{large}$ (200M/453M, 0.380), as shown in Figure 1. When compared with dense models, PatchMoE$_{base}$ (200M/453M) achieves comparable average MAE (0.335) on LTF benchmarks with fewer activation parameters, such as Sundial$_{large}$ (444M, 0.338) and Moirai$_{large}$(311M, 0.349). These improvements simply prove that our proposed architecture—featuring a novel *patch-wise experts* and *sample-wise router*—is essential for effectively specializing experts and attaining superior generalization across all domains.

**Full-shot Forecasting.** After just one epoch of finetuning, PatchMoE demonstrates remarkable performance, consistently outperforming all baselines to establish new state-of-the-art results across both LTF benchmarks and our newly proposed datasets, as shown in Table 3. Compared to outstanding classical models such as TiDE and PatchTST, PatchMoE$_{ultra}$ achieves an average of 20% MAE improvement on our proposed datasets. It also secures the top rank in 50 out of 60 test configurations, spanning six LTF benchmarks and four prediction lengths, while outperforming other advanced MoE-based (MoLE, FreqMoE) and multi-patches (Pathformer, MTST) models. Notably, while Pathformer is also an adaptive routing model, our hierarchical MoE architecture, trained on large-scale time series data, achieves improvements by a large margin.

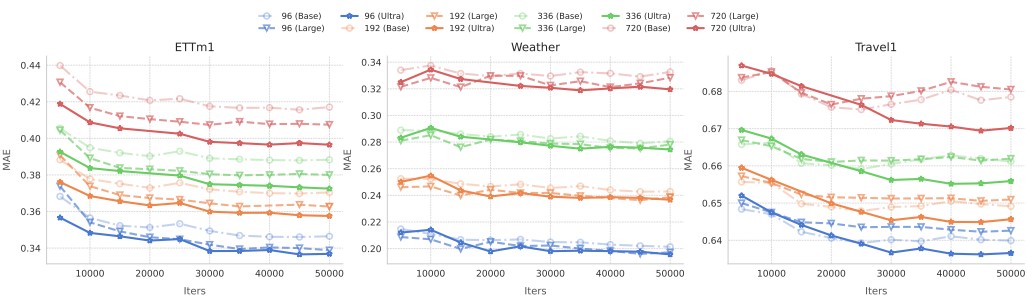

Figure 5: Test MAE against training iterations (a proxy for data volume) on three representative datasets. The results show strong scalability: larger models (Ultra > Large > Base) consistently improve performance (**parameter scalability**), and performance improves with more data (**data scalability**). This synergistic gain is most pronounced on challenging long-horizon tasks.

## 4.3 SCALABILITY & EFFICIENCY ANALYSIS

**Data & Parameter Scalability.** We investigate the scalability of PatchMoE with respect to parameters, data utilization, and task complexity. Figure 5 illustrates the results on three representative datasets. Firstly, for any given task, larger models consistently yield lower error, confirming strong parameter scalability. Secondly, the steady performance improvement with more training iterations highlights the effective data scalability of our model. Critically, these scaling benefits are synergistic and most pronounced on difficult, long-horizon forecasting tasks, e.g., the gap between curves for the 720-step forecast is wider than for the 96-step forecast.

**Training Efficiency.** As shown in Figure 3 and Table 9, our training framework demonstrates superior training efficiency. For PatchMoE$_{ultra}$, this framework outperforms standard PyTorch methods like FSDP, DDP and DP by over $5.2\times$ and $3.0\times$, $6.3\times$, respectively. When comparing with models of a similar scale, PatchMoE$_{base}$ not only trains approximately **3×** faster, but also achieves a substantially better performance than Time-MoE$_{large}$(453M). This highlights the significant efficiency advantages of our training framework.

## 4.4 ABLATION STUDIES

**PatchMoE Architecture Analysis.** To validate the effective of each key module in our proposed PatchMoE, we conduct extensive ablation studies (see Table 4 and Appendix D.4.1) by selectively removing or replacing modules, which are grouped into several main categories: MoE Architecture, Patch-wise Experts, Sample-wise Hierarchical and Pre-training Framework. The results unequivocally demonstrate that each component is integral to the overall performance of the model.

Table 4: **Ablation study of PatchMoE$_{ultra}$.** We evaluate the impact of removing or replacing key modules on the zero-shot performance on LTF benchmarks and our proposed datasets.

| | | Our proposed Datasets | | | | LTF Benchmarks | | | |
| --- | --- | --- | --- | --- | --- | --- | --- | --- | --- |
| | | MSE | | MAE | | MSE | | MAE | |
| | **PatchMoE$_{ultra}$** | **1.203** | | **0.521** | | **0.290** | | **0.326** | |
| **MoE Architecture** | *w/o* Mixture-of-Experts | 1.235 | ↓ **2.66%** | 0.527 | ↓ **1.09%** | 0.303 | ↓ **4.55%** | 0.332 | ↓ **1.84%** |
| | *w/o* Load-balance Auxiliary Loss | 1.221 | ↓ **1.44%** | 0.525 | ↓ **0.70%** | 0.292 | ↓ **0.92%** | 0.326 | - |
| **Patch-wise Experts** | *w/o* Patch-wise Experts | 1.210 | ↓ **0.58%** | 0.519 | - | 0.302 | ↓ **4.15%** | 0.332 | ↓ **1.89%** |
| | *w/o* Multi-Layer Expert | 1.231 | ↓ **2.27%** | 0.526 | ↓ **0.96%** | 0.298 | ↓ **2.94%** | 0.334 | ↓ **2.40%** |
| **Sample-wise Hierarchical Router** | *w/o* Sample-wise Router | 1.240 | ↓ **3.02%** | 0.540 | ↓ **3.58%** | 0.294 | ↓ **1.44%** | 0.327 | ↓ **0.31%** |
| | *w/o* Hierarchical Modeing | 1.212 | ↓ **0.75%** | 0.528 | ↓ **1.41%** | 0.295 | ↓ **1.90%** | 0.331 | ↓ **1.53%** |
| | *w/o* Doubly Residual Stacking | 1.203 | - | 0.519 | - | 0.321 | ↓ **10.77%** | 0.353 | ↓ **8.34%** |
| **Pre-training Framework** | *w/o* Input Mask | 1.200 | - | 0.522 | ↓ **0.26%** | 0.293 | ↓ **1.04%** | 0.328 | ↓ **0.67%** |
| | *w/o* Multi-Resolution Loss | 1.216 | ↓ **1.02%** | 0.522 | ↓ **0.19%** | 0.295 | ↓ **1.73%** | 0.329 | ↓ **0.86%** |

**Model Design Analysis.** We conducted a series of analysis to understand the impact of key design choices of PatchMoE$_{ultra}$, as shown in Figure 6, to demonstrate the impact of model sparsity, type of Patch Expert, training loss function and type of prediction head, respectively. See Appendix D.4.2 for detailed analysis.

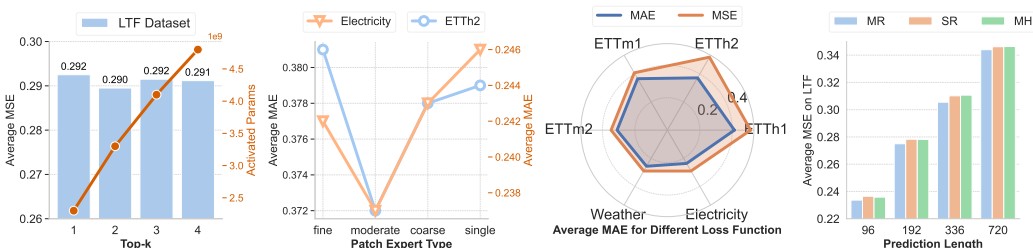

Figure 6: **Analysis of Key Model Components**. (**1**) Impact of model sparsity (controlled by Top-$k$) on performance and computational cost (Activated Params). (**2**) Effect of different *expert group* patch granularity combinations. (**3**) Performance comparison of models trained with MAE loss vs. MSE loss on LTF benchmarks. (**4**) Comparison of three prediction head types (**M**ulti-**R**esolution(**MR**), **S**ingle-**R**esolution(**SR**), **M**ulti-**H**ead(**MH**)) across different prediction lengths.

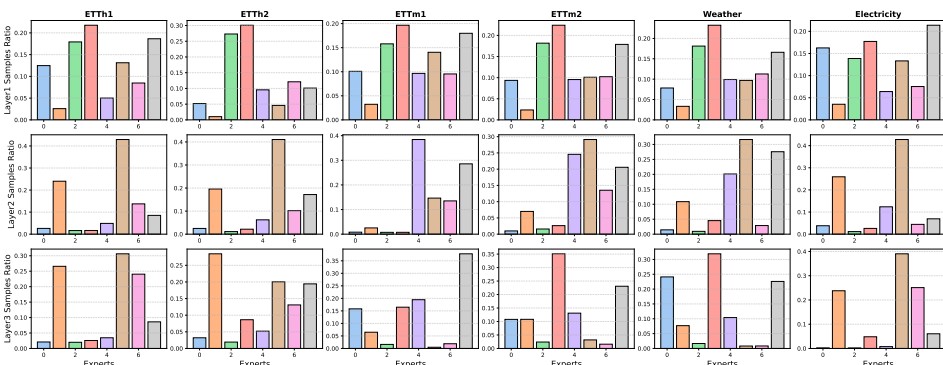

Figure 7: **Routing visualization of expert allocation** on LTF benchmarks. Each bar chart illustrates the proportion of samples for which a given specialized expert is selected by Sample-wise Top-k Router. Visualization of more datasets and prediction horizons can be found in Appendix D.5.

## 4.5 ROUTING VISUALIZATION

The Sample-wise Hierarchical Router achieves strong performance through expert specialization, routing samples to a small, targeted subset of experts instead of uniform usage (visualized in Figure 7, 9-16). This specialization is characterized by two main properties: Hierarchical Refinement, where routing becomes more focused in deeper layers to handle finer residual signals, and Dataset-Adaptive Specialization, where expert roles dynamically adjust based on each dataset's unique characteristics. This dynamic routing enables powerful conditional computation, allowing the model to efficiently model diverse temporal patterns by dispatching each sample to its most suitable experts, thereby driving its superior performance.

## 5 CONCLUSION

In this paper, we propose PatchMoE, a time series foundation model with hierarchical MoE architecture. To tackle diverse *inter*-series and *intra*-series patterns in large-scale datasets, we introduce Patch-wise Experts and a Sample-wise Hierarchical Router, respectively. Furthermore, we develop a highly efficient training framework for the time series modality based on Megatron-LM that implements expert parallelism. This framework enables us to scale the parameters of a time series foundation model to **8.5**B for the first time, achieving a $3\times$ to $5\times$ improvement in training efficiency. Extensive experiments and ablation studies demonstrate that PatchMoE achieves significant improvements in zero-shot and full-shot settings and validate its novel components. This work lays the foundation for future parameter scaling and MoE design in time series domain while providing the community with a vital and efficient pre-training framework.

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

# A    DETAILS OF METHODOLOGY

## A.1    PATCH-WISE EXPERTS NETWORK

We provide the details of calculation pipeline of our Patch-Wise Experts here. Each expert is a stack of $L_{\text{Expert}}$ identical Transformer Vaswani et al. (2017) layers, and is distinguished by a specialized tokenizer with a unique patch size. In the following description, we omit the MoE layer notation $l$ for ease of reading.

**Patch Embedding.** Given a specific patch size of $P_i$ of the $i$-th expert, where $i \in [1, N]$, the input backcast $\boldsymbol{b} \in \mathbb{R}^L$ is first segmented into $N_i$ non-overlapping patches, where $N_i = \lfloor L/P_i \rfloor$. The patchified input $\boldsymbol{P} \in \mathbb{R}^{N_i \times P_i}$, is processed by a Multi-Layer Perceptron (MLP) with a SwiGLU activation function Shazeer (2020) to generate the patch embeddings:

$$\boldsymbol{E}_j = (\text{Swish}(\boldsymbol{P}\boldsymbol{W}_1 \odot (\boldsymbol{P}\boldsymbol{W}_2)))\boldsymbol{W}_3, \tag{6}$$

where $\boldsymbol{W}_{1,2} \in \mathbb{R}^{P_i \times d_{\text{ff}}}$, $\boldsymbol{W}_3 \in \mathbb{R}^{d_{\text{ff}} \times d_{\text{model}}}$, $\odot$ denotes element-wise multiplication, and $\text{Swish}(\boldsymbol{x}) = \boldsymbol{x} \cdot \sigma(\boldsymbol{x})$ with $\sigma$ being the sigmoid function. Meanwhile, the patch mask is constructed according to the Input Mask method 3.3. Specifically, a patch is entirely masked if it contains any masked time points. We enforce the last patch to keep unmasked, so that at least one patch is valid. The patch embeddings $\boldsymbol{E}_i \in \mathbb{R}^{N_i \times d_{\text{model}}}$ and the corresponding patch masks serve as the input of the Transformer block.

**Transformer Block.** The Transformer block is a stack of $L_{\text{Expert}}$ Transformer layers. Each layer consists of a Multi-Head Self-Attention (MHSA) and a Feed-Forward Network (FFN) module, and adapts Pre-LN Touvron et al. (2023) with RMSNorm Zhang & Sennrich (2019) by residual connections. We omit the Transformer layer index for simplicity here. Let $\boldsymbol{Z} \in \mathbb{R}^{N_i \times d_{\text{model}}}$ be the output of the previous Transformer layer, which is first normalized and is then utilized to compute the query, key, and value matrices:

$$\boldsymbol{Z}' = \text{RMSNorm}(\boldsymbol{Z}), \quad \boldsymbol{Q} = \boldsymbol{Z}'\boldsymbol{W}_q, \boldsymbol{K} = \boldsymbol{Z}'\boldsymbol{W}_k, \boldsymbol{V} = \boldsymbol{Z}'\boldsymbol{W}_v. \tag{7}$$

Let $f(\cdot, j)$ denote the application of the RoPE Su et al. (2024) for position $j$, the attention output is:

$$\boldsymbol{A} = \text{Softmax}(\frac{f(\boldsymbol{Q}, j)f(\boldsymbol{K}, j)}{\sqrt{d_k}} + \boldsymbol{M})\boldsymbol{V}, \quad \boldsymbol{Z}_{\text{mid}} = \boldsymbol{Z} + \text{MHSA}(\boldsymbol{A}), \tag{8}$$

where $d_k$ is the hidden size of each head, and $\boldsymbol{M} \in \mathbb{R}^{N_i \times N_i}$ is derived from the patch mask by setting the corresponding masked position to $-\infty$. The intermediate representation $\boldsymbol{Z}_{\text{mid}} \in \mathbb{R}^{N_i \times d_{\text{model}}}$ is then normalized and passed through the FFN:

$$\boldsymbol{Z}'_{\text{mid}} = \text{RMSNorm}(\boldsymbol{Z}_{\text{mid}}), \quad \boldsymbol{Z}_{\text{ffn}} = \text{FFN}(\boldsymbol{Z}'_{\text{mid}}). \tag{9}$$

The output of each Transformer layer is formed by residual connection:

$$\boldsymbol{Z}_{\text{output}} = \boldsymbol{Z}_{\text{mid}} + \boldsymbol{Z}_{\text{ffn}} \in \mathbb{R}^{N_i \times d_{\text{model}}} \tag{10}$$

**Flatten Head.** After $L_{\text{Expert}}$ layers, the output of the last Transformer layer $\boldsymbol{Z}_{\text{output}}$ undergoes a final normalization, and is then flattened and projected by two separate linear layers with $\boldsymbol{W}_{\text{backcast}} \in \mathbb{R}^{N_i \cdot d_{\text{model}} \times L}$, $\boldsymbol{W}_{\text{forecast}} \in \mathbb{R}^{N_i \cdot d_{\text{model}} \times T}$ to produce the backcast and forecast outputs of the expert:

$$\boldsymbol{z} = \text{Flatten}(\text{RMSNorm}(\boldsymbol{Z}^{(L_{\text{Expert}})})), \quad \Delta\mathbf{b}_i = \boldsymbol{z}\boldsymbol{W}_{\text{backcast}}, \quad \Delta\mathbf{f}_i = \boldsymbol{z}\boldsymbol{W}_{\text{forecast}}. \tag{11}$$

## A.2 DOMAIN-DIVERSITY SAMPLING ON LARGE-SCALE DATASETS.

We employ a domain-diversity sampling strategy with pre-defined weights to mitigate domain imbalance in the time series datasets. Specifically, the total number of samples $S$ for the pre-training is first determined by the number of training iterations $I$ and the global batch size $B$, i.e. $S = I * B$. Subsequently, given a large-scale dataset of $Q$ distinct domains $\mathcal{D} = \{D_i\}_{i=1}^{Q}$, each domain $D_i$ is assigned a pre-defined sampling weight $w_i$, where $\Sigma_{i=1}^{N} w_i = 1$. This weight dictates that the domain contributes $S \times w_i$ samples to the overall mixed pre-training data. Assuming domain $D_i$ comprises $M_i$ time series and contains a total of $S_i$ potential samples, we first shuffle the order of $M_i$ time series. Let $L$ be the maximum input length of the model, and $T$ be the prediction length. We then adopt the following sampling rules to deal with time series with varying length:

- For time series with a length at least $L + T$: We extract samples using a sliding window with a stride of $\lfloor S \times w_i / S_i \rfloor$.

- For time series with a length less than $L + T$ but at least $T$: We take the last $T$ time points as the label. The remaining preceding time points are left-padded to a length of $L$ to form the input sequence.

- For time series with a length shorter than $T$: These series are discarded as they are insufficient to form a complete label.

This composite strategy ensures sample diversity of domain $D_i$ through stride sampling, while also maximizing the inclusion of shorter time series in our pre-training dataset. Finally, all samples are shuffled globally.

## A.3 INFERENCE STRATEGY

The design of our multi-resolution loss 3.2, which operates over a set of distinct sub-horizons $\{T_1, \ldots, T_H\}$, inspires us to design an *Coarse-to-Fine* inference strategy to produce outputs of arbitrary length. The core idea is to prioritize the use of the largest sub-horizon $T_H$ at each step, leveraging its superior long-range forecasting capability, and then iteratively use smaller horizons to fill the remaining length.

# B IMPLEMENTATION DETAILS

## B.1 DATA STORAGE AND LOADING

We pre-train PatchMoE on large-scale datasets comprising over 300 billion time points Xiaoming et al. (2025); Liu et al. (2024b); Woo et al. (2024). For each constituent dataset, we store the data as a continuous binary file (*.bin*) alongside a corresponding metadata file (*.idx*). This setup allows us to leverage memory mapping based on the metadata, loading only the required data segments into the address space on-the-fly rather than pre-loading all files into memory. This high-throughput data loading strategy is crucial for our sampling method detailed in Appendix A.2. To ensure a fair evaluation and prevent data leakage, we explicitly excluded all datasets used for zero-shot evaluation from our pre-training corpus. Specifically, neither the binary data files (*.bin*) nor the metadata index files (*.idx*) contain any information of zero-shot evaluation datasets.

## B.2 ADDITIONAL EXPERIMENTAL SETTINGS

**Training Configuration.** Leveraging our efficient training framework, all three variants of Patch-MoE are pre-trained for 50,000 iterations with a global batch size of 4096, consuming over 200 million samples and spanning about 590 billion time points. We train the model using the AdamW optimizer Loshchilov & Hutter (2019) with $\beta_1 = 0.9$, $\beta_2 = 0.95$, and a weight decay of 0.1. The learning rate schedule consists of a linear warmup for the first 0.1% of training steps to a peak value of $6 \times 10^{-5}$, followed by a cosine decay to a minimum of $6 \times 10^{-6}$. To improve training efficiency, we employ `bf16` precision and set the expert parallelism size at 4.

**Mask Padding Configuration.** We use a padding value of 255 for both the Input Mask method 3.3 and the masking of short time series as described in Appendix A.2. Since the time series data is normalized to a standard normal distribution ($\mathcal{N}(0, 1)$) prior to masking, the value 255 is highly unlikely to collide with any valid value of unmasked time points. Furthermore, this integer value is well within the representation range of the `bf16` data type. This masking scheme greatly facilitates the sample dispatching by our Sample-wise Router. Upon receiving a dispatched sample, each Patch-wise Expert can identify the masked positions by detecting the value 255, which in turn enables the generation of the correct attention mask as detailed in Appendix A.1.

**Patch-wise Experts Configuration.** The patch sizes for the Patch-wise Experts are configured differently for each model variant. We set $\{36, 48, 96, 120\}$ for PatchMoE$_{base}$, $\{36, 64, 96, 120\}$ for PatchMoE$_{large}$, and $\{16, 24, 36, 48, 64, 72, 96, 120\}$ for PatchMoE$_{ultra}$, respectively. The patch size for the Shared Expert is set to 32 for all variants.

## C DATASET STATISTICS

We conduct extensive experiments on LTF benchmarks, including the ETT (ETTh1, ETTh2, ETTm1, ETTm2) Zhou et al. (2021), Weather[3] and Electricity[4] dataset. Additionally, to further test model generalization on more volatile, human-centric processes, we propose three real-world datasets from the high-value commercial domains, including Travel1, Travel2 and E-comm. These datasets, drawn from the tourism and e-commerce sectors with recent data from 2023-2024, provide a crucial supplement to existing benchmarks for evaluating real-world forecasting capabilities. A detailed description of each dataset is provided in Table 5.

Table 5: Detailed descriptions of LTF benchmarks and our proposed datasets.

| Task | Domain | Date Range | Test Range | Dataset | Time Series | Total Time Stamps | Test Time Stamps | Frequency |
|---|---|---|---|---|---|---|---|---|
| **Long-Term Forecasting Benchmark Wu et al. (2021)** | Temperature | 2016-07-01-2018-06-26 | 2017-10-24-2018-06-26 | ETTm1 ETTm2 | 7 | 69,680 | 11,520 | 15 mins |
| | | | | ETTh1 ETTh2 | 7 | 17,420 | 2,880 | 1 hour |
| | Weather | 2020-01-01-2021-01-01 | 2020-10-19-2021-01-01 | Weather | 21 | 52,696 | 10,539 | 10 mins |
| | Electricity | 2016-07-01-2019-07-02 | 2018-11-24-2019-07-02 | Electricity | 321 | 26,304 | 5,260 | 1 hour |
| **Our Proposed Datasets** | Tourism | 2023-06-04-2023-12-20 | 2023-10-02-2023-12-20 | Travel1 | 3 | 4,776 | 1,896 | 1 hour |
| | Tourism E-commerce | 2024-04-03-2024-12-30 | 2024-08-01-2024-12-30 | Travel2 E-comm | 5 5 | 6,528 6,528 | 3,648 3,648 | 1 hour 1 hour |

Table 6: Detailed zero-shot performance of TSFMs on LTF benchmarks and our proposed datasets. A lower MSE or MAE indicates a better prediction. The subscripts $b$, $l$, and $u$ denote base, large, and ultra, respectively. Results of datasets included in pre-training procedure are denoted by a dash($-$). Results of unreleased models that cannot be evaluated on our proposed datasets are denoted by a slash(\). **Red**: the best, Blue: the 2nd best.

| Models | | Ours | | | | | | MoE-based models | | | | | | Dense baselines | | | | | | | | | | | |
|---|---|---|---|---|---|---|---|---|---|---|---|---|---|---|---|---|---|---|---|---|---|---|---|---|---|
| | | PatchMoE$_b$ | | PatchMoE$_l$ | | PatchMoE$_u$ | | Time-MoE$_l$ | | Time-MoE$_u$ | | Moirai-MoE$_b$ | | Sundial$_b$ | | Sundial$_l$ | | Timer-XL | | Moirai$_l$ | | Chronos$_l$ | | TimesFM | |
| Metrics | | MSE | MAE | MSE | MAE | MSE | MAE | MSE | MAE | MSE | MAE | MSE | MAE | MSE | MAE | MSE | MAE | MSE | MAE | MSE | MAE | MSE | MAE | MSE | MAE |
| ETTh1 | 96 | 0.361 | 0.381 | 0.372 | 0.382 | 0.366 | **0.378** | 0.350 | 0.382 | 0.349 | 0.379 | 0.412 | 0.388 | 0.348 | 0.385 | **0.346** | 0.383 | 0.369 | 0.391 | 0.381 | 0.388 | 0.441 | 0.390 | 0.414 | 0.404 |
| | 192 | 0.406 | 0.407 | 0.418 | 0.410 | 0.412 | **0.405** | 0.388 | 0.412 | 0.395 | 0.413 | 0.471 | 0.422 | 0.393 | 0.418 | **0.386** | 0.410 | 0.405 | 0.413 | 0.434 | 0.415 | 0.502 | 0.424 | 0.465 | 0.434 |
| | 336 | 0.438 | 0.423 | 0.455 | 0.427 | 0.438 | **0.420** | 0.411 | 0.430 | 0.417 | 0.430 | 0.544 | 0.449 | 0.422 | 0.440 | **0.410** | 0.426 | 0.418 | 0.423 | 0.495 | 0.445 | 0.576 | 0.467 | 0.503 | 0.456 |
| | 720 | 0.452 | 0.453 | 0.477 | 0.451 | 0.463 | 0.447 | 0.427 | 0.455 | 0.457 | 0.462 | 0.601 | 0.487 | 0.481 | 0.493 | 0.438 | 0.459 | **0.423** | **0.441** | 0.611 | 0.510 | 0.835 | 0.583 | 0.511 | 0.481 |
| | Avg. | 0.414 | 0.416 | 0.430 | 0.418 | 0.420 | **0.413** | **0.394** | 0.420 | 0.404 | 0.421 | 0.507 | 0.436 | 0.411 | 0.434 | 0.395 | 0.420 | 0.404 | 0.417 | 0.480 | 0.440 | 0.588 | 0.466 | 0.473 | 0.444 |
| ETTh2 | 96 | 0.283 | 0.327 | 0.282 | 0.327 | 0.289 | **0.326** | 0.302 | 0.354 | 0.292 | 0.352 | 0.312 | 0.342 | 0.271 | 0.333 | **0.269** | 0.330 | 0.283 | 0.342 | 0.296 | 0.330 | 0.320 | 0.345 | 0.315 | 0.349 |
| | 192 | 0.343 | 0.369 | 0.336 | **0.366** | 0.348 | 0.367 | 0.364 | 0.385 | 0.347 | 0.379 | 0.372 | 0.384 | 0.327 | 0.376 | **0.325** | 0.373 | 0.340 | 0.379 | 0.361 | 0.371 | 0.406 | 0.399 | 0.388 | 0.395 |
| | 336 | 0.367 | 0.391 | 0.364 | 0.392 | 0.372 | **0.389** | 0.417 | 0.425 | 0.406 | 0.419 | 0.409 | 0.416 | **0.354** | 0.402 | **0.354** | 0.400 | 0.366 | 0.400 | 0.390 | 0.390 | 0.492 | 0.453 | 0.422 | 0.427 |
| | 720 | 0.381 | 0.416 | 0.380 | 0.416 | **0.376** | **0.406** | 0.537 | 0.496 | 0.439 | 0.447 | 0.448 | 0.446 | 0.381 | 0.435 | 0.389 | 0.443 | 0.397 | 0.431 | 0.423 | 0.418 | 0.603 | 0.511 | 0.443 | 0.454 |
| | Avg. | 0.344 | 0.376 | 0.340 | 0.375 | 0.346 | **0.372** | 0.405 | 0.415 | 0.371 | 0.399 | 0.385 | 0.397 | **0.333** | 0.387 | 0.334 | 0.387 | 0.347 | 0.388 | 0.368 | 0.377 | 0.455 | 0.427 | 0.392 | 0.406 |
| ETTm1 | 96 | 0.323 | 0.346 | 0.301 | 0.339 | 0.294 | 0.337 | 0.309 | 0.357 | 0.281 | 0.341 | 0.384 | 0.376 | 0.280 | 0.334 | **0.273** | **0.329** | 0.317 | 0.356 | 0.380 | 0.361 | 0.457 | 0.403 | 0.361 | 0.370 |
| | 192 | 0.360 | 0.370 | 0.335 | 0.363 | 0.322 | 0.358 | 0.346 | 0.381 | **0.305** | 0.358 | 0.463 | 0.469 | 0.321 | 0.366 | 0.312 | **0.357** | 0.358 | 0.381 | 0.412 | 0.383 | 0.530 | 0.450 | 0.414 | 0.405 |
| | 336 | 0.386 | 0.388 | 0.357 | 0.380 | **0.341** | **0.373** | 0.373 | 0.408 | 0.369 | 0.395 | 0.441 | 0.455 | 0.350 | 0.389 | 0.343 | 0.378 | 0.386 | 0.401 | 0.436 | 0.400 | 0.577 | 0.481 | 0.445 | 0.429 |
| | 720 | 0.438 | 0.417 | 0.396 | 0.407 | **0.372** | **0.397** | 0.475 | 0.477 | 0.469 | 0.472 | 0.555 | 0.556 | 0.394 | 0.418 | 0.397 | 0.413 | 0.430 | 0.431 | 0.462 | 0.420 | 0.660 | 0.526 | 0.512 | 0.471 |
| | Avg. | 0.377 | 0.380 | 0.347 | 0.372 | 0.332 | **0.366** | 0.376 | 0.406 | 0.356 | 0.392 | 0.461 | 0.464 | 0.336 | 0.377 | **0.331** | 0.369 | 0.373 | 0.392 | 0.422 | 0.391 | 0.556 | 0.465 | 0.433 | 0.419 |
| ETTm2 | 96 | 0.185 | 0.260 | 0.172 | 0.249 | 0.172 | **0.248** | 0.197 | 0.286 | 0.198 | 0.288 | 0.218 | 0.283 | **0.170** | 0.256 | 0.172 | 0.255 | 0.189 | 0.277 | 0.211 | 0.274 | 0.197 | 0.271 | 0.202 | 0.270 |
| | 192 | 0.253 | 0.307 | 0.228 | **0.289** | 0.231 | 0.290 | 0.250 | 0.322 | 0.235 | 0.312 | 0.281 | 0.327 | 0.229 | 0.300 | **0.227** | 0.296 | 0.241 | 0.315 | 0.281 | 0.318 | 0.254 | 0.314 | 0.289 | 0.321 |
| | 336 | 0.309 | 0.346 | 0.278 | **0.323** | 0.279 | 0.325 | 0.337 | 0.375 | 0.293 | 0.348 | 0.398 | 0.369 | 0.281 | 0.337 | **0.275** | 0.331 | 0.286 | 0.348 | 0.341 | 0.355 | 0.313 | 0.353 | 0.360 | 0.366 |
| | 720 | 0.375 | 0.389 | 0.352 | **0.376** | 0.352 | 0.377 | 0.480 | 0.461 | 0.427 | 0.428 | 0.456 | 0.430 | 0.351 | 0.387 | **0.343** | 0.378 | 0.375 | 0.402 | 0.485 | 0.428 | 0.416 | 0.415 | 0.462 | 0.430 |
| | Avg. | 0.280 | 0.326 | 0.258 | **0.309** | 0.258 | 0.310 | 0.316 | 0.361 | 0.288 | 0.344 | 0.338 | 0.352 | 0.258 | 0.320 | **0.254** | 0.315 | 0.273 | 0.336 | 0.330 | 0.344 | 0.295 | 0.338 | 0.328 | 0.347 |
| Weather | 96 | 0.164 | 0.201 | 0.159 | 0.197 | **0.157** | **0.196** | 0.159 | 0.213 | **0.157** | 0.211 | 0.233 | 0.233 | **0.157** | 0.205 | **0.157** | 0.208 | 0.171 | 0.225 | 0.199 | 0.211 | 0.194 | 0.235 | - | - |
| | 192 | 0.207 | 0.242 | 0.202 | 0.239 | **0.199** | **0.237** | 0.215 | 0.266 | 0.208 | 0.256 | 0.256 | 0.273 | 0.205 | 0.251 | 0.207 | 0.256 | 0.221 | 0.271 | 0.246 | 0.251 | 0.249 | 0.285 | - | - |
| | 336 | 0.257 | 0.281 | 0.256 | 0.278 | **0.251** | **0.274** | 0.291 | 0.322 | 0.255 | 0.290 | 0.311 | 0.310 | 0.253 | 0.289 | 0.259 | 0.295 | 0.274 | 0.311 | 0.274 | 0.291 | 0.302 | 0.327 | - | - |
| | 720 | 0.335 | 0.333 | 0.328 | 0.328 | **0.314** | **0.320** | 0.415 | 0.400 | 0.405 | 0.397 | 0.347 | 0.352 | 0.320 | 0.336 | 0.327 | 0.342 | 0.356 | 0.370 | 0.337 | 0.340 | 0.372 | 0.378 | - | - |
| | Avg. | 0.241 | 0.264 | 0.236 | 0.260 | **0.230** | **0.257** | 0.270 | 0.300 | 0.256 | 0.288 | 0.287 | 0.292 | 0.234 | 0.270 | 0.238 | 0.275 | 0.256 | 0.294 | 0.264 | 0.273 | 0.279 | 0.306 | - | - |
| ECL | 96 | 0.130 | 0.221 | 0.128 | 0.216 | **0.124** | **0.211** | - | - | - | - | 0.141 | 0.224 | 0.132 | 0.229 | 0.130 | 0.227 | 0.141 | 0.237 | 0.153 | 0.241 | 0.152 | 0.229 | - | - |
| | 192 | 0.146 | 0.236 | 0.142 | 0.230 | **0.138** | **0.225** | - | - | - | - | 0.166 | 0.245 | 0.152 | 0.250 | 0.150 | 0.247 | 0.159 | 0.254 | 0.169 | 0.255 | 0.172 | 0.250 | - | - |
| | 336 | 0.163 | 0.253 | 0.158 | 0.246 | **0.152** | **0.240** | - | - | - | - | 0.188 | 0.270 | 0.173 | 0.271 | 0.170 | 0.268 | 0.177 | 0.272 | 0.187 | 0.273 | 0.203 | 0.276 | - | - |
| | 720 | 0.206 | 0.291 | 0.194 | 0.278 | **0.187** | **0.272** | - | - | - | - | 0.257 | 0.323 | 0.218 | 0.311 | 0.214 | 0.307 | 0.219 | 0.308 | 0.237 | 0.313 | 0.289 | 0.337 | - | - |
| | Avg. | 0.161 | 0.250 | 0.156 | 0.242 | **0.150** | **0.237** | - | - | - | - | 0.188 | 0.266 | 0.169 | 0.265 | 0.166 | 0.262 | 0.174 | 0.268 | 0.186 | 0.270 | 0.204 | 0.273 | - | - |
| 1st Cnt | | 0 | | 5 | | **35** | | 1 | | 2 | | 0 | | 4 | | 15 | | 2 | | 0 | | 0 | | 0 | |
| 2nd Cnt | | 5 | | **28** | | 10 | | 4 | | 2 | | 0 | | 11 | | 7 | | 1 | | 1 | | 0 | | 0 | |
| Travel1 | 96 | 0.867 | 0.643 | 0.863 | 0.640 | **0.852** | **0.637** | 0.977 | 0.674 | \ | \ | 0.879 | 0.668 | 0.861 | 0.648 | \ | \ | 0.902 | 0.677 | 1.914 | 1.056 | 1.293 | 0.763 | 0.865 | 0.646 |
| | 192 | 0.898 | 0.652 | 0.892 | 0.649 | **0.880** | **0.646** | 1.083 | 0.701 | \ | \ | 0.896 | 0.669 | 0.897 | 0.658 | \ | \ | 0.935 | 0.689 | 1.981 | 1.089 | 1.381 | 0.792 | 0.883 | 0.654 |
| | 336 | 0.941 | 0.663 | 0.935 | 0.661 | **0.916** | **0.656** | 1.144 | 0.718 | \ | \ | 0.920 | 0.672 | 0.951 | 0.671 | \ | \ | 1.007 | 0.722 | 2.011 | 1.094 | 1.614 | 0.852 | 0.918 | 0.668 |
| | 720 | 1.009 | 0.680 | 0.994 | 0.678 | 0.972 | **0.670** | 1.223 | 0.739 | \ | \ | 1.043 | 0.696 | 1.045 | 0.690 | \ | \ | 1.136 | 0.777 | 2.025 | 1.091 | 2.644 | 1.116 | **0.968** | 0.683 |
| | Avg. | 0.929 | 0.660 | 0.921 | 0.657 | **0.905** | **0.652** | 1.107 | 0.708 | \ | \ | 0.934 | 0.676 | 0.938 | 0.667 | \ | \ | 0.995 | 0.716 | 1.983 | 1.082 | 1.733 | 0.881 | 0.908 | 0.663 |
| Travel2 | 96 | 0.225 | **0.251** | 0.227 | 0.252 | 0.231 | 0.254 | **0.218** | 0.258 | \ | \ | 0.250 | 0.260 | 0.231 | 0.260 | \ | \ | 0.250 | 0.279 | 1.053 | 0.739 | 0.271 | 0.271 | 0.235 | 0.255 |
| | 192 | 0.253 | **0.262** | 0.255 | 0.265 | 0.257 | 0.266 | **0.248** | 0.273 | \ | \ | 0.275 | 0.269 | 0.263 | 0.274 | \ | \ | 0.297 | 0.299 | 1.129 | 0.765 | 0.291 | 0.283 | 0.266 | 0.268 |
| | 336 | 0.285 | **0.273** | 0.282 | 0.277 | **0.278** | 0.278 | 0.281 | 0.296 | \ | \ | 0.301 | 0.286 | 0.307 | 0.290 | \ | \ | 0.398 | 0.344 | 1.136 | 0.767 | 0.336 | 0.286 | 0.308 | 0.286 |
| | 720 | 0.339 | 0.302 | 0.340 | 0.305 | **0.314** | **0.298** | 0.383 | 0.377 | \ | \ | 0.402 | 0.331 | 0.402 | 0.327 | \ | \ | 0.623 | 0.431 | 1.163 | 0.733 | 0.344 | 0.306 | 0.332 | 0.332 |
| | Avg. | 0.275 | **0.272** | 0.276 | 0.275 | **0.270** | 0.274 | 0.283 | 0.301 | \ | \ | 0.307 | 0.286 | 0.301 | 0.288 | \ | \ | 0.392 | 0.338 | 1.120 | 0.751 | 0.310 | 0.287 | 0.285 | 0.285 |
| E-comm | 96 | 2.226 | 0.566 | 2.186 | 0.576 | **2.143** | **0.560** | 2.542 | 0.639 | \ | \ | 2.590 | 0.614 | 2.245 | 0.612 | \ | \ | 2.307 | 0.651 | 4.224 | 1.314 | 2.183 | 0.575 | 2.343 | 0.595 |
| | 192 | 2.433 | 0.636 | 2.356 | 0.634 | **2.309** | **0.617** | 2.801 | 0.712 | \ | \ | 2.859 | 0.668 | 2.432 | 0.678 | \ | \ | 2.455 | 0.701 | 4.662 | 1.401 | 2.359 | 0.626 | 2.518 | 0.669 |
| | 336 | 2.623 | 0.683 | 2.491 | 0.665 | **2.471** | **0.656** | 3.096 | 0.777 | \ | \ | 3.093 | 0.739 | 2.624 | 0.719 | \ | \ | 2.827 | 0.814 | 4.769 | 1.412 | 2.750 | 0.757 | 2.727 | 0.712 |
| | 720 | 2.911 | 0.733 | 2.831 | 0.716 | **2.818** | **0.713** | 3.709 | 0.916 | \ | \ | 3.067 | 0.769 | 2.910 | 0.751 | \ | \ | 3.080 | 0.839 | 4.827 | 1.398 | 2.969 | 0.831 | 3.072 | 0.769 |
| | Avg. | 2.548 | 0.654 | 2.466 | 0.648 | **2.435** | **0.637** | 3.037 | 0.761 | \ | \ | 2.902 | 0.698 | 2.553 | 0.690 | \ | \ | 2.667 | 0.751 | 4.620 | 1.381 | 2.565 | 0.697 | 2.665 | 0.686 |
| 1st Cnt | | 4 | | 0 | | **23** | | 2 | | \ | | 0 | | 0 | | \ | | 0 | | 0 | | 0 | | 1 | |
| 2nd Cnt | | 6 | | **15** | | 2 | | 1 | | \ | | 0 | | 1 | | \ | | 0 | | 0 | | 1 | | 4 | |

# D    ADDITIONAL RESULTS

## D.1    ZERO-SHOT RESULTS.

Table 6 provides details of zero-shot performance of four prediction horizons $\{96, 192, 336, 720\}$, on LTF benchmarks and our proposed datasets. Following Sundial Liu et al. (2025e), the look back window length is set as 2880.

We compare our model against recent prominent Time Series Foundation Models (TSFMs). Due to space constraints, our main comparison in Table 6 focuses on the largest available version of each model. A comprehensive average performance across all model versions is presented in Figure 1. The evaluation results were obtained as follows:

- **Results from Published Papers:** For LTF benchmarks, most metrics are directly adopted from the officially published papers.
  - Results for $Sundial_{small}$, $Sundial_{base}$[5], $Sundial_{large}$ (2025e), Timer-XL (2025d), $Time\text{-}MoE_{base}$[6], $Time\text{-}MoE_{large}$[7], $Time\text{-}MoE_{ultra}$ (2025), and TimesFM[8] (2024) are sourced from the Sundial paper (2025e).
  - Results for $Moirai_{small}$[9], $Moirai_{base}$[10], $Moirai_{large}$[11] (2024), $Chronos_{small}$[12], $Chronos_{base}$[13], $Chronos_{large}$[14] (2024), and Moment (2024) are sourced from the Time-MoE paper (2025).

- **Results from Our Evaluation:** For models where specific benchmark results were not available, we ran our own evaluation.
  - We evaluated the officially released checkpoints of $Moirai\text{-}MoE_{small}$[15], $Moirai\text{-}MoE_{base}$[16] (2025b), and Timer[17] (2024b) using a standardized evaluation pipeline.

- **Results on Our Proposed Datasets:** All competing models were evaluated using their official checkpoints. We employed an identical data loading and evaluation procedure for all models to ensure a fair comparison.

## D.2    FULL-SHOT RESULTS

Table 7 demonstrates detailed full-shot results of four prediction horizons $\{96, 192, 336, 720\}$, on LTF benchmarks and our proposed datasets. The base, large and ultra variants of PatchMoE were fine-tuned for one epoch using a context length of 2880. For all domain-specific baselines, we report the better of two values: the performance metric reported in their official paper and the result obtained from our evaluation using an input length of 2880. Table 7 presents a selection of nine representative baselines, categorized into Classic Models, Multi-patches Models, and MoE-based Models. A more comprehensive comparison of average MAE across a wider range of domain models is illustrated in Figure 8. The evaluation results were obtained as follows:

- **Results on LTF Benchmarks:** All the results are taken from the publicly available papers for each baseline. We categorize them into three groups based on their model architectures and specify the input length on which the reported results were obtained.

---

[3] https://www.bgc-jena.mpg.de/wetter/

[4] https://archive.ics.uci.edu/dataset/321/electricityloaddiagrams20112014

[5] $Sundial_{base}$ (128M): https://huggingface.co/thuml/sundial-base-128m

[6] $Time\text{-}MoE_{base}$(453M-A200M): https://huggingface.co/Maple728/TimeMoE-50M

[7] $Time\text{-}MoE_{large}$(453M-A200M): https://huggingface.co/Maple728/TimeMoE-200M

[8] TimesFM (200M): https://huggingface.co/google/timesfm-1.0-200m-pytorch

[9] $Moirai_{small}$(14M): https://huggingface.co/Salesforce/moirai-1.0-R-small

[10] $Moirai_{base}$(91M):https://huggingface.co/Salesforce/moirai-1.0-R-base

[11] $Moirai_{large}$(311M): https://huggingface.co/Salesforce/moirai-1.0-R-large

[12] $chronos_{small}$(17M): https://huggingface.co/amazon/chronos-t5-small

[13] $chronos_{base}$(200M): https://huggingface.co/amazon/chronos-t5-base

[14] $chronos_{large}$(710M): https://huggingface.co/amazon/chronos-t5-large

[15] $Moirai\text{-}MoE_{small}$(117M-A11M): https://huggingface.co/Salesforce/moirai-moe-1.0-R-small

[16] $Moirai\text{-}MoE_{base}$(935M-86M): https://huggingface.co/Salesforce/moirai-moe-1.0-R-base

[17] Timer(84M): https://huggingface.co/thuml/timer-base-84m

Table 7: Detailed full-shot performance of PatchMoE and domain models on well-acknowledge datasets and our proposed datasets. A lower MSE or MAE indicates a better prediction. **Red**: the best, Blue: the 2nd best.

| Models | | Ours | | | | | | Classic Models | | | | | | | | | | Multi-patches Models | | | | MoE-based Models | | | |
|---|---|---|---|---|---|---|---|---|---|---|---|---|---|---|---|---|---|---|---|---|---|---|---|---|---|---|
| | | PatchMoE$_b$ | | PatchMoE$_l$ | | PatchMoE$_u$ | | PatchTST | | iTransformer | | TiDE | | DLinear | | TimeMixer | | Pathformer | | MTST | | MoLE | | FreqMoE | |
| Metrics | | MSE | MAE | MSE | MAE | MSE | MAE | MSE | MAE | MSE | MAE | MSE | MAE | MSE | MAE | MSE | MAE | MSE | MAE | MSE | MAE | MSE | MAE | MSE | MAE |
| ETTh1 | 96 | 0.351 | 0.375 | 0.346 | 0.372 | 0.346 | 0.370 | 0.370 | 0.400 | 0.386 | 0.405 | 0.375 | 0.398 | 0.370 | 0.399 | 0.375 | 0.400 | 0.382 | 0.400 | 0.376 | 0.393 | 0.381 | 0.400 | 0.371 | 0.388 |
| | 192 | 0.400 | 0.403 | 0.389 | 0.400 | 0.387 | 0.395 | 0.413 | 0.429 | 0.441 | 0.436 | 0.412 | 0.422 | 0.405 | 0.416 | 0.436 | 0.429 | 0.440 | 0.427 | 0.429 | 0.422 | 0.429 | 0.433 | 0.426 | 0.422 |
| | 336 | 0.435 | 0.420 | 0.419 | 0.435 | 0.414 | 0.409 | 0.422 | 0.440 | 0.487 | 0.458 | 0.435 | 0.433 | 0.439 | 0.443 | 0.484 | 0.458 | 0.454 | 0.432 | 0.444 | 0.436 | 0.453 | 0.445 | 0.475 | 0.447 |
| | 720 | 0.445 | 0.448 | 0.425 | 0.446 | 0.424 | 0.433 | 0.447 | 0.468 | 0.503 | 0.491 | 0.454 | 0.465 | 0.472 | 0.490 | 0.498 | 0.482 | 0.479 | 0.461 | 0.469 | 0.466 | 0.505 | 0.493 | 0.488 | 0.459 |
| | Avg. | 0.408 | 0.412 | 0.395 | 0.413 | 0.393 | 0.402 | 0.413 | 0.434 | 0.454 | 0.447 | 0.419 | 0.430 | 0.422 | 0.437 | 0.448 | 0.442 | 0.541 | 0.507 | 0.430 | 0.429 | 0.442 | 0.443 | 0.440 | 0.429 |
| ETTh2 | 96 | 0.273 | 0.319 | 0.275 | 0.323 | 0.266 | 0.321 | 0.274 | 0.337 | 0.297 | 0.349 | 0.270 | 0.336 | 0.289 | 0.353 | 0.289 | 0.341 | 0.279 | 0.331 | 0.276 | 0.333 | 0.322 | 0.371 | 0.287 | 0.337 |
| | 192 | 0.340 | 0.362 | 0.335 | 0.364 | 0.321 | 0.359 | 0.341 | 0.382 | 0.380 | 0.400 | 0.332 | 0.380 | 0.383 | 0.418 | 0.372 | 0.392 | 0.349 | 0.380 | 0.353 | 0.382 | 0.361 | 0.404 | 0.361 | 0.386 |
| | 336 | 0.361 | 0.381 | 0.363 | 0.390 | 0.347 | 0.379 | 0.329 | 0.384 | 0.428 | 0.432 | 0.360 | 0.407 | 0.448 | 0.465 | 0.386 | 0.414 | 0.348 | 0.382 | 0.357 | 0.395 | 0.382 | 0.423 | 0.407 | 0.423 |
| | 720 | 0.371 | 0.404 | 0.375 | 0.408 | 0.365 | 0.405 | 0.379 | 0.422 | 0.427 | 0.445 | 0.419 | 0.451 | 0.605 | 0.551 | 0.412 | 0.434 | 0.398 | 0.424 | 0.406 | 0.430 | 0.442 | 0.459 | 0.414 | 0.438 |
| | Avg. | 0.336 | 0.367 | 0.337 | 0.371 | 0.325 | 0.366 | 0.331 | 0.381 | 0.383 | 0.407 | 0.345 | 0.394 | 0.431 | 0.447 | 0.365 | 0.395 | 0.344 | 0.379 | 0.348 | 0.385 | 0.377 | 0.414 | 0.367 | 0.396 |
| ETTml | 96 | 0.290 | 0.331 | 0.291 | 0.331 | 0.285 | 0.337 | 0.293 | 0.346 | 0.334 | 0.368 | 0.306 | 0.349 | 0.299 | 0.343 | 0.320 | 0.357 | 0.316 | 0.346 | 0.323 | 0.360 | 0.296 | 0.341 | 0.314 | 0.356 |
| | 192 | 0.329 | 0.356 | 0.313 | 0.351 | 0.314 | 0.356 | 0.333 | 0.370 | 0.377 | 0.391 | 0.335 | 0.366 | 0.335 | 0.365 | 0.361 | 0.381 | 0.366 | 0.370 | 0.363 | 0.386 | 0.338 | 0.365 | 0.356 | 0.380 |
| | 336 | 0.356 | 0.374 | 0.338 | 0.373 | 0.337 | 0.372 | 0.369 | 0.392 | 0.426 | 0.420 | 0.364 | 0.384 | 0.369 | 0.386 | 0.390 | 0.404 | 0.386 | 0.394 | 0.393 | 0.406 | 0.370 | 0.391 | 0.385 | 0.404 |
| | 720 | 0.404 | 0.403 | 0.382 | 0.397 | 0.372 | 0.397 | 0.416 | 0.420 | 0.491 | 0.459 | 0.413 | 0.413 | 0.425 | 0.421 | 0.454 | 0.441 | 0.460 | 0.432 | 0.453 | 0.441 | 0.424 | 0.419 | 0.446 | 0.445 |
| | Avg. | 0.345 | 0.366 | 0.331 | 0.363 | 0.327 | 0.366 | 0.353 | 0.382 | 0.407 | 0.410 | 0.355 | 0.378 | 0.357 | 0.379 | 0.381 | 0.396 | 0.382 | 0.386 | 0.383 | 0.398 | 0.357 | 0.379 | 0.375 | 0.396 |
| ETTm2 | 96 | 0.162 | 0.244 | 0.159 | 0.243 | 0.162 | 0.242 | 0.166 | 0.256 | 0.180 | 0.264 | 0.161 | 0.251 | 0.167 | 0.260 | 0.175 | 0.258 | 0.170 | 0.248 | 0.174 | 0.256 | 0.165 | 0.254 | 0.173 | 0.266 |
| | 192 | 0.217 | 0.284 | 0.219 | 0.283 | 0.214 | 0.279 | 0.223 | 0.296 | 0.250 | 0.309 | 0.215 | 0.289 | 0.224 | 0.303 | 0.237 | 0.299 | 0.238 | 0.295 | 0.243 | 0.302 | 0.235 | 0.298 | 0.235 | 0.310 |
| | 336 | 0.270 | 0.319 | 0.269 | 0.318 | 0.264 | 0.312 | 0.274 | 0.329 | 0.311 | 0.348 | 0.267 | 0.326 | 0.281 | 0.342 | 0.298 | 0.340 | 0.293 | 0.331 | 0.301 | 0.340 | 0.282 | 0.335 | 0.290 | 0.350 |
| | 720 | 0.338 | 0.370 | 0.348 | 0.372 | 0.346 | 0.369 | 0.362 | 0.385 | 0.412 | 0.407 | 0.352 | 0.383 | 0.397 | 0.421 | 0.391 | 0.396 | 0.390 | 0.389 | 0.397 | 0.395 | 0.369 | 0.386 | 0.385 | 0.424 |
| | Avg. | 0.247 | 0.304 | 0.249 | 0.304 | 0.247 | 0.301 | 0.256 | 0.317 | 0.288 | 0.332 | 0.249 | 0.312 | 0.267 | 0.332 | 0.275 | 0.323 | 0.273 | 0.316 | 0.279 | 0.323 | 0.263 | 0.318 | 0.271 | 0.338 |
| Weather | 96 | 0.157 | 0.199 | 0.154 | 0.196 | 0.149 | 0.195 | 0.149 | 0.198 | 0.174 | 0.214 | 0.149 | 0.198 | 0.176 | 0.237 | 0.163 | 0.209 | 0.156 | 0.192 | 0.175 | 0.216 | 0.161 | 0.210 | 0.168 | 0.215 |
| | 192 | 0.200 | 0.240 | 0.195 | 0.237 | 0.192 | 0.236 | 0.194 | 0.241 | 0.221 | 0.254 | 0.194 | 0.241 | 0.220 | 0.282 | 0.208 | 0.250 | 0.206 | 0.240 | 0.219 | 0.255 | 0.199 | 0.248 | 0.212 | 0.253 |
| | 336 | 0.244 | 0.274 | 0.241 | 0.271 | 0.237 | 0.271 | 0.245 | 0.282 | 0.278 | 0.296 | 0.245 | 0.282 | 0.265 | 0.319 | 0.251 | 0.287 | 0.254 | 0.282 | 0.276 | 0.296 | 0.255 | 0.291 | 0.268 | 0.291 |
| | 720 | 0.315 | 0.325 | 0.304 | 0.317 | 0.296 | 0.314 | 0.314 | 0.334 | 0.358 | 0.347 | 0.314 | 0.334 | 0.323 | 0.362 | 0.339 | 0.341 | 0.340 | 0.336 | 0.351 | 0.346 | 0.328 | 0.341 | 0.342 | 0.345 |
| | Avg. | 0.229 | 0.260 | 0.224 | 0.255 | 0.219 | 0.254 | 0.226 | 0.264 | 0.257 | 0.278 | 0.226 | 0.264 | 0.246 | 0.300 | 0.240 | 0.271 | 0.239 | 0.263 | 0.255 | 0.278 | 0.236 | 0.273 | 0.248 | 0.276 |
| ECL | 96 | 0.127 | 0.218 | 0.125 | 0.214 | 0.123 | 0.212 | 0.129 | 0.222 | 0.148 | 0.240 | 0.129 | 0.220 | 0.140 | 0.237 | 0.153 | 0.259 | 0.145 | 0.236 | 0.160 | 0.248 | 0.144 | 0.239 | 0.152 | 0.246 |
| | 192 | 0.144 | 0.234 | 0.142 | 0.230 | 0.141 | 0.229 | 0.147 | 0.240 | 0.162 | 0.253 | 0.147 | 0.240 | 0.153 | 0.249 | 0.166 | 0.256 | 0.167 | 0.256 | 0.171 | 0.263 | 0.166 | 0.260 | 0.165 | 0.255 |
| | 336 | 0.159 | 0.249 | 0.156 | 0.245 | 0.156 | 0.245 | 0.163 | 0.259 | 0.178 | 0.269 | 0.163 | 0.259 | 0.169 | 0.267 | 0.185 | 0.277 | 0.186 | 0.275 | 0.188 | 0.281 | 0.178 | 0.273 | 0.181 | 0.274 |
| | 720 | 0.195 | 0.282 | 0.188 | 0.275 | 0.189 | 0.275 | 0.197 | 0.290 | 0.225 | 0.317 | 0.197 | 0.290 | 0.203 | 0.301 | 0.225 | 0.310 | 0.231 | 0.309 | 0.230 | 0.315 | 0.198 | 0.291 | 0.219 | 0.307 |
| | Avg. | 0.156 | 0.246 | 0.153 | 0.241 | 0.152 | 0.240 | 0.159 | 0.253 | 0.178 | 0.270 | 0.159 | 0.252 | 0.166 | 0.264 | 0.182 | 0.275 | 0.182 | 0.269 | 0.187 | 0.277 | 0.172 | 0.266 | 0.179 | 0.271 |
| 1st Cnt | | 4 | | 12 | | 50 | | 2 | | 0 | | 1 | | 0 | | 0 | | 1 | | 0 | | 0 | | 0 | |
| 2nd Cnt | | 21 | | 28 | | 9 | | 1 | | 0 | | 6 | | 0 | | 0 | | 0 | | 0 | | 0 | | 0 | |
| Travel1 | 96 | 0.867 | 0.643 | 0.863 | 0.640 | 0.852 | 0.637 | 1.803 | 1.026 | 1.523 | 0.927 | 1.976 | 1.076 | 1.512 | 0.941 | 4.464 | 1.620 | 0.912 | 0.664 | 1.304 | 0.853 | 1.137 | 0.781 | 1.559 | 0.961 |
| | 192 | 0.898 | 0.652 | 0.892 | 0.649 | 0.880 | 0.646 | 1.282 | 0.818 | 1.574 | 0.927 | 1.732 | 0.993 | 1.144 | 0.754 | 1.899 | 1.061 | 1.046 | 0.744 | 1.155 | 0.782 | 0.960 | 0.698 | 1.525 | 0.916 |
| | 336 | 0.941 | 0.663 | 0.935 | 0.661 | 0.916 | 0.656 | 1.333 | 0.801 | 1.636 | 0.925 | 1.785 | 1.007 | 1.165 | 0.744 | 2.913 | 1.333 | 0.975 | 0.682 | 1.174 | 0.784 | 1.021 | 0.715 | 1.854 | 1.006 |
| | 720 | 1.009 | 0.680 | 0.994 | 0.678 | 0.972 | 0.670 | 1.449 | 0.830 | 1.960 | 1.022 | 1.795 | 1.000 | 1.260 | 0.762 | 4.250 | 1.590 | 1.147 | 0.779 | 1.209 | 0.787 | 1.117 | 0.743 | 1.948 | 1.021 |
| | Avg. | 0.929 | 0.660 | 0.921 | 0.657 | 0.905 | 0.652 | 1.467 | 0.869 | 1.673 | 0.950 | 1.822 | 1.019 | 1.270 | 0.800 | 3.382 | 1.401 | 1.020 | 0.717 | 1.211 | 0.802 | 1.059 | 0.734 | 1.722 | 0.976 |
| Travel2 | 96 | 0.219 | 0.249 | 0.221 | 0.248 | 0.215 | 0.247 | 0.287 | 0.317 | 0.278 | 0.303 | 0.411 | 0.399 | 0.299 | 0.322 | 0.503 | 0.458 | 0.226 | 0.254 | 0.236 | 0.269 | 0.242 | 0.282 | 0.593 | 0.512 |
| | 192 | 0.248 | 0.260 | 0.244 | 0.258 | 0.237 | 0.258 | 0.296 | 0.316 | 0.311 | 0.320 | 0.345 | 0.353 | 0.264 | 0.287 | 0.283 | 0.303 | 0.251 | 0.263 | 0.263 | 0.280 | 0.254 | 0.275 | 0.439 | 0.418 |
| | 336 | 0.274 | 0.271 | 0.261 | 0.266 | 0.259 | 0.269 | 0.302 | 0.314 | 0.337 | 0.333 | 0.377 | 0.363 | 0.279 | 0.290 | 0.323 | 0.328 | 0.281 | 0.274 | 0.296 | 0.295 | 0.286 | 0.286 | 0.872 | 0.670 |
| | 720 | 0.321 | 0.300 | 0.293 | 0.284 | 0.305 | 0.292 | 0.347 | 0.335 | 0.375 | 0.359 | 0.435 | 0.386 | 0.329 | 0.313 | 0.434 | 0.396 | 0.333 | 0.295 | 0.355 | 0.321 | 0.352 | 0.316 | 0.452 | 0.430 |
| | Avg. | 0.266 | 0.270 | 0.255 | 0.264 | 0.254 | 0.267 | 0.308 | 0.321 | 0.325 | 0.329 | 0.392 | 0.375 | 0.293 | 0.303 | 0.386 | 0.371 | 0.273 | 0.272 | 0.288 | 0.291 | 0.284 | 0.290 | 0.589 | 0.508 |
| E-comm | 96 | 2.143 | 0.538 | 2.100 | 0.547 | 2.100 | 0.556 | 2.296 | 0.696 | 2.545 | 0.749 | 3.491 | 1.059 | 2.509 | 0.808 | 5.826 | 1.407 | 2.340 | 0.614 | 2.180 | 0.618 | 2.279 | 0.669 | 3.508 | 1.119 |
| | 192 | 2.297 | 0.595 | 2.264 | 0.603 | 2.264 | 0.612 | 2.391 | 0.718 | 2.691 | 0.804 | 2.929 | 0.860 | 2.322 | 0.693 | 2.507 | 0.743 | 2.400 | 0.655 | 2.327 | 0.676 | 2.374 | 0.689 | 3.523 | 1.146 |
| | 336 | 2.454 | 0.633 | 2.404 | 0.637 | 2.422 | 0.653 | 2.565 | 0.760 | 2.722 | 0.812 | 3.143 | 0.899 | 2.439 | 0.715 | 3.191 | 0.902 | 2.596 | 0.728 | 2.533 | 0.731 | 2.577 | 0.739 | 4.246 | 1.273 |
| | 720 | 2.755 | 0.695 | 2.702 | 0.689 | 2.716 | 0.702 | 2.867 | 0.784 | 3.185 | 0.896 | 3.387 | 0.908 | 2.735 | 0.742 | 4.236 | 1.065 | 2.891 | 0.752 | 2.895 | 0.795 | 2.857 | 0.768 | 4.139 | 1.254 |
| | Avg. | 2.412 | 0.615 | 2.368 | 0.619 | 2.376 | 0.631 | 2.530 | 0.740 | 2.786 | 0.815 | 3.238 | 0.932 | 2.501 | 0.740 | 3.940 | 1.029 | 2.557 | 0.687 | 2.484 | 0.705 | 2.522 | 0.716 | 3.854 | 1.198 |
| 1st Cnt | | 4 | | 11 | | 18 | | 0 | | 0 | | 0 | | 0 | | 0 | | 0 | | 0 | | 0 | | 0 | |
| 2nd Cnt | | 5 | | 18 | | 7 | | 0 | | 0 | | 0 | | 0 | | 0 | | 0 | | 0 | | 0 | | 0 | |

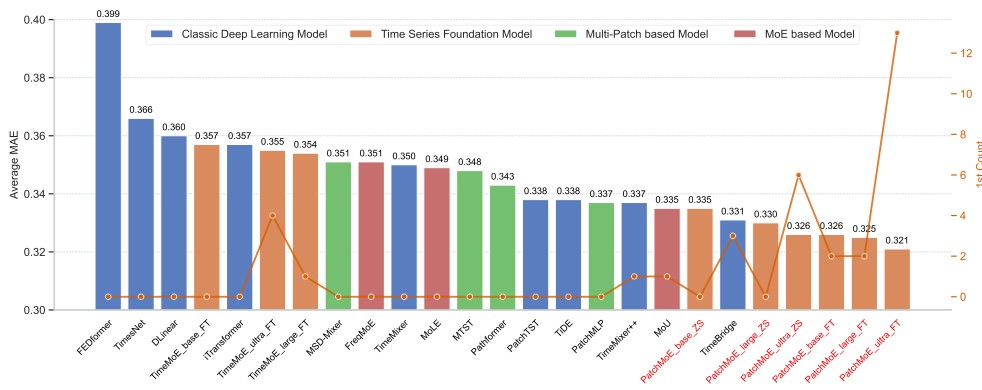

Figure 8: Patch-MoE achieves state-of-the-art full-shot forecasting on the long-term forecasting datasets (LTF) Wu et al. (2021) benchmark, surpassing all up-to-date models which categorized as three group-Classic Deep Learning Model, Multi-Patches based Model and MoE based Model.

- **Classic Models.** We choose some state-of-the-art models to serve as baselines, including FED-former[18] (2022) (input length is 96, results except for ETT are taken from iTransformer), PatchTST[19] (input length is 512) (2023), TimesNet[20] (input length is 96) (2023), DLinear[21] (2023) (input length is 336), TiDE[22] (2023) (input length is 720), TimeMixer[23] (2024) (input length are selected from {96, 192, 336, 512}), iTransformer[24] (2024a) (input length is 96), TimeMixer++[25] (2025a) (input length is 96), TimeBridge[26] (2025a) (input length are selected from {96, 192, 336, 512, 720}).

- **Multi-patches Based Models.** Pathformer[27] (2024) (input length is 96), MTST[28] (2024b) (input length is 336), MSD-Mixer[29] (2024) (input length is 96), PatchMLP[30] (2025) (input length is 96, and only the average results are released; the details are not disclosed.).

- **MoE-based Models.** FreqMoE[31] (2025) (input length is 96), MoLE[32] (2024) (input length are selected from {96, 192, 336, 720}, and the results are taken from MoU), MoU[33] (2025) (input length are selected from {96, 192, 336, 720}).

- **Results on Our Proposed Datasets:** All baseline models were evaluated using their official implementation code and all parameters follow the original design. We employed an identical data loading and evaluation procedure for all models to ensure a fair comparison.

Furthermore, we present a comparison with Time-MoE of the full-shot setting in Table 8, where PatchMoE consistently achieves superior performance, particularly with the ultra version. This superiority still holds at comparable parameter scales. Specifically, our PatchMoE$_{large}$ (1.2B / 2.5B) achieves an overall average MSE/MAE of 0.307/0.341, surpassing the similarly-sized Time-MoE$_{ultra}$ (1.1B / 2.4B) which scores 0.313/0.355. Likewise, PatchMoE-base (200M / 440M) with metrics of

---

[18]FEDformer: https://github.com/MAZiqing/FEDformer

[19]PatchTST: https://github.com/yuqinie98/PatchTST

[20]TimesNet: https://github.com/thuml/TimesNet

[21]DLinear: https://github.com/vivva/DLinear

[22]TiDE: https://github.com/zihanghliu/TiDE

[23]TimeMixer: https://github.com/kwuking/TimeMixer

[24]iTransformer: https://github.com/thuml/iTransformer

[25]TimeMixer++: https://github.com/kwuking/TimeMixer

[26]TimeBridge: https://github.com/Hank0626/TimeBridge

[27]Pathformer: https://github.com/decisionintelligence/pathformer

[28]MTST: https://github.com/networkslab/MTST

[29]MSD-Mixer: https://github.com/zshhans/MSD-Mixer

[30]PatchMLP: https://github.com/TangPeiwang/PatchMLP

[31]FreqMoE: https://github.com/sunbus100/FreqMoE-main

[32]MoLE:https://github.com/SBiswas03/Project-on-Mixture-of-Linear-Experts-for-Long-term-Time-Series-forecasting-

[33]MoU: https://github.com/lunaaa95/mou

0.313/0.342 is superior to Time-MoE-large (200M / 453M) at 0.316/0.354. These results strongly validate the effectiveness of our novel MoE architectural design.

Table 8: Full shot results compared with Time-MoE. **Red**: the best, Blue:the 2nd best.

| Models | | PatchMoE (Ours) | | | | | | Time-MoE | | | | | |
|---|---|---|---|---|---|---|---|---|---|---|---|---|---|
| | | PatchMoE$_{base}$ | | PatchMoE$_{large}$ | | PatchMoE$_{ultra}$ | | Time-MoE$_{base}$ | | Time-MoE$_{large}$ | | Time-MoE$_{ultra}$ | |
| Metrics | | MSE | MAE | MSE | MAE | MSE | MAE | MSE | MAE | MSE | MAE | MSE | MAE |
| ETTh1 | 96 | 0.351 | 0.375 | 0.346 | 0.372 | 0.346 | 0.370 | 0.345 | 0.373 | 0.335 | 0.371 | 0.323 | 0.365 |
| | 192 | 0.400 | 0.403 | 0.389 | 0.400 | 0.387 | 0.395 | 0.372 | 0.396 | 0.374 | 0.400 | 0.359 | 0.391 |
| | 336 | 0.435 | 0.420 | 0.419 | 0.435 | 0.414 | 0.409 | 0.389 | 0.412 | 0.390 | 0.412 | 0.388 | 0.418 |
| | 720 | 0.445 | 0.448 | 0.425 | 0.446 | 0.424 | 0.433 | 0.410 | 0.443 | 0.402 | 0.433 | 0.425 | 0.450 |
| | Avg. | 0.408 | 0.412 | 0.395 | 0.413 | 0.393 | 0.402 | 0.379 | 0.406 | 0.375 | 0.404 | 0.374 | 0.406 |
| ETTh2 | 96 | 0.273 | 0.319 | 0.275 | 0.323 | 0.266 | 0.321 | 0.276 | 0.340 | 0.278 | 0.335 | 0.274 | 0.338 |
| | 192 | 0.340 | 0.362 | 0.335 | 0.364 | 0.321 | 0.359 | 0.331 | 0.371 | 0.345 | 0.373 | 0.330 | 0.370 |
| | 336 | 0.361 | 0.381 | 0.363 | 0.390 | 0.347 | 0.379 | 0.373 | 0.402 | 0.384 | 0.402 | 0.362 | 0.396 |
| | 720 | 0.371 | 0.404 | 0.375 | 0.408 | 0.365 | 0.405 | 0.404 | 0.431 | 0.437 | 0.437 | 0.370 | 0.417 |
| | Avg. | 0.336 | 0.367 | 0.337 | 0.371 | 0.325 | 0.366 | 0.346 | 0.386 | 0.361 | 0.387 | 0.334 | 0.380 |
| ETTm1 | 96 | 0.290 | 0.331 | 0.291 | 0.331 | 0.285 | 0.337 | 0.286 | 0.334 | 0.264 | 0.325 | 0.256 | 0.323 |
| | 192 | 0.329 | 0.356 | 0.313 | 0.351 | 0.314 | 0.356 | 0.307 | 0.358 | 0.295 | 0.350 | 0.281 | 0.343 |
| | 336 | 0.356 | 0.374 | 0.338 | 0.373 | 0.337 | 0.372 | 0.354 | 0.390 | 0.323 | 0.376 | 0.326 | 0.374 |
| | 720 | 0.404 | 0.403 | 0.382 | 0.397 | 0.372 | 0.397 | 0.433 | 0.445 | 0.409 | 0.435 | 0.454 | 0.452 |
| | Avg. | 0.345 | 0.366 | 0.331 | 0.363 | 0.327 | 0.366 | 0.345 | 0.382 | 0.323 | 0.372 | 0.329 | 0.373 |
| ETTm2 | 96 | 0.162 | 0.244 | 0.159 | 0.243 | 0.162 | 0.242 | 0.172 | 0.265 | 0.169 | 0.259 | 0.183 | 0.273 |
| | 192 | 0.217 | 0.284 | 0.219 | 0.283 | 0.214 | 0.279 | 0.228 | 0.306 | 0.223 | 0.295 | 0.223 | 0.301 |
| | 336 | 0.270 | 0.319 | 0.269 | 0.318 | 0.264 | 0.312 | 0.281 | 0.345 | 0.293 | 0.341 | 0.278 | 0.339 |
| | 720 | 0.338 | 0.370 | 0.348 | 0.372 | 0.346 | 0.369 | 0.403 | 0.424 | 0.451 | 0.433 | 0.425 | 0.424 |
| | Avg. | 0.247 | 0.304 | 0.249 | 0.304 | 0.247 | 0.301 | 0.271 | 0.335 | 0.284 | 0.332 | 0.277 | 0.334 |
| Weather | 96 | 0.157 | 0.199 | 0.154 | 0.196 | 0.149 | 0.195 | 0.151 | 0.203 | 0.149 | 0.201 | 0.154 | 0.208 |
| | 192 | 0.200 | 0.240 | 0.195 | 0.237 | 0.192 | 0.236 | 0.195 | 0.246 | 0.192 | 0.244 | 0.202 | 0.251 |
| | 336 | 0.244 | 0.274 | 0.241 | 0.271 | 0.237 | 0.271 | 0.247 | 0.288 | 0.245 | 0.285 | 0.252 | 0.287 |
| | 720 | 0.315 | 0.325 | 0.304 | 0.317 | 0.296 | 0.314 | 0.352 | 0.366 | 0.352 | 0.365 | 0.392 | 0.376 |
| | Avg. | 0.229 | 0.260 | 0.224 | 0.255 | 0.219 | 0.254 | 0.236 | 0.276 | 0.235 | 0.274 | 0.250 | 0.281 |
| Average | | 0.313 | 0.342 | 0.307 | 0.341 | 0.302 | 0.338 | 0.315 | 0.357 | 0.316 | 0.354 | 0.313 | 0.355 |

Table 9: Comparison of training efficiency across various different training frameworks and models on $8\times$ NVIDIA H200-141GB GPUs.

| Model | Activated / Total Parameters | Training Framework | Batch Size | Speed (s/iter) |
|---|---|---|---|---|
| PatchMoE$_{ultra}$ | 3.8B / 8.5B | Megatron-LM | 256 | **0.260** |
| | | FSDP[1] | 256 | 1.363 |
| | | DDP[2] | 256 | 0.787 |
| | | DP[3] | 128 | 1.635 |
| PatchMoE$_{large}$ | 1.2B / 2.5B | Megatron-LM | 256 | **0.165** |
| PatchMoE$_{base}$ | 200M / 440M | Megatron-LM | 256 | **0.126** |
| [*]Time-MoE$_{large}$ | 200M / 453M | transformers.Trainer | 4 | 0.461 |
| [*]Time-MoE$_{base}$ | 50M / 113M | transformers.Trainer | 4 | 0.363 |

[1] FSDP: `torch.distributed.fsdp.FullyShardedDataParallel`.   [2] DDP: `torch.nn.parallel.DistributedDataParallel`.
[3] DP: `torch.nn.DataParallel`.   [*] For Time-MoE models, increasing the micro-batch size beyond 4 resulted in an out-of-memory (OOM) error.

## D.3 TRAINING EFFICIENCY

Table 9 presents the detailed numerical settings corresponding to the efficiency comparison illustrated in Figure 3. The results highlight the superior training efficiency of three versions of our PatchMoE variants. Compared to alternative distributed PyTorch frameworks and other models on comparable or even smaller scales, our approach achieves a training speed that is at least $3\times$ faster.

### D.4 ABLATION STUDIES

#### D.4.1 PATCHMOE ARCHITECTURE ANALYSIS

To thoroughly validate the effectiveness of each component of PatchMoE architecture, we conduct comprehensive ablation studies on the PatchMoE$_{ultra}$ model. We systematically remove or replace key modules and evaluate their impact on zero-shot forecasting performance across both LTF benchmarks and our proposed datasets. As shown in Table 4, the results clearly demonstrate that each design choice makes an indispensable contribution to the overall performance of PatchMoE.

**Effectiveness of MoE Architecture.** We begin by validating the core value of the MoE architecture. In the *w/o Mixture-of-Experts* experiment, we replace the model with a *dense* version having a comparable number of parameters, specifically implemented as a 12-layer PatchTST model with a fixed patch size of 32. The results show that the performance of this dense model degrades significantly, with a 4.55% increase in MSE on LTF benchmarks, demonstrating the superiority of MoE in handling diverse *intra*-series patterns through expert specialization. Furthermore, we remove the *Load-balance Auxiliary Loss* by setting its coefficient to zero. The resulting performance drop (e.g., a 1.44% MSE increase on our proposed datasets) underscores the criticality of this loss in preventing router collapse and ensuring balanced expert training.

**Impact of Patch-wise Experts.** A key innovation in PatchMoE is its expert design. To validate the effectiveness of *Patch-wise Experts*, we conduct an experiment where all experts share a fixed patch size of 32, instead of each having a specialized patch tokenizer. The significant performance decline (a 4.15% MSE increase on LTF benchmarks) confirms that equipping experts with varying patch scales is key to capturing *inter*-series diversity. Next, in the *w/o Multi-Layer Expert* experiment, we simplify each expert from a multi-layer Transformer stack to a single layer, and to maintain the model's total depth, we increase the number of routing steps to 12. The noticeable drop in performance (a 2.94% MSE increase on LTF benchmarks) indicates that providing sufficient depth within each expert to process complex patterns is more effective than more frequent routing to shallow experts.

**Analysis of Hierarchical Modeling and Routing.** We analyze the routing and modeling mechanisms within our hierarchical architecture. First, to verify the necessity of the *Sample-wise Router*, we replace it with the conventional *token-wise* routing, where experts are reverted to standard FFNs. The sharp drop in performance (a 3.02% and 3.58% increase in MSE and MAE on our proposed datasets, respectively) highlights that for time series, routing the entire sample as a coherent unit is crucial for preserving pattern integrity. Second, in the *w/o Hierarchical Modeling* experiment, we use only a single MoE layer (one routing step) and increase the expert depth to 12 layers. The performance degradation (a 1.90% MSE increase on LTF benchmarks) demonstrates that the hierarchical approach of progressively decomposing the signal across multiple layers is superior to a single, monolithic processing step. Finally, we examine the *Doubly Residual Stacking*. We disable the backcast residual subtraction and forecast aggregation, relying only on the final layer's output for prediction. This results in the most severe performance collapse across all ablations (a staggering 10.77% increase in MSE on LTF benchmarks), unequivocally proving that this mechanism is the cornerstone for effective signal decomposition and accurate forecasting.

**Analysis of the Pre-training Framework.** Lastly, we assess the impact of our pre-training strategies. In the *w/o Input Mask* experiment, the model is trained only on complete samples padded to the maximum length of 2880. Although the impact is relatively smaller, a consistent performance drop is observed (a 1.04% MSE increase on LTF benchmarks), suggesting that enabling the model to handle variable-length inputs enhances its generalization capability. Similarly, removing the *Multi-Resolution Loss* and calculating the loss only on the full prediction horizon (336 steps) also leads to a performance decline (a 1.73% MSE increase on LTF benchmarks). This confirms that supervising the model across multiple sub-horizons helps it generate more robust and accurate predictions across different time scales.

#### D.4.2 MODEL DESIGN ANALYSIS

We conducted a series of analysis to understand the impact of key design choices of PatchMoE$_{ultra}$, as shown in Figure 6, to demonstrate the impact of model sparsity, type of Patch Expert, training loss function and type of prediction head, respectively.

**Impact of Model Sparsity.** As shown in the figure, we analyze the impact of model sparsity by varying Top-$k$ from 1 to 4. While the number of activated parameters increases approximately linearly with $k$, he prediction accuracy (lower average MSE) does not improve monotonically. The best performance is achieved at $k = 2$ with an average MSE of 0.290. Increasing $k$ further to 3 and 4 leads to a slight degradation in accuracy at a higher computational cost. This highlights the trade-off between performance and efficiency, justifying our choice of $k = 2$ as the optimal configuration for our ultra version.

**Impact of Patch-wise Experts Type.** The center-left panel shows that a moderate granularity for patch experts (i.e. the list of patch-wise experts in each MoE layer is $[16, 24, 36, 48, 64, 72, 96, 120]$) achieves the lowest average MAE compared to finer (i.e. $[16, 18, 20, 24, 30, 32, 36, 48]$), coarser (i.e. $[16, 48, 72, 96, 120, 144, 180, 240]$, or single-patch approaches (i.e. $32$). This suggests that a moderate level of specialization is optimal for capturing temporal patterns.

**Impact of Training Loss Function.** The radar chart (center-right) clearly shows that the model trained with MAE loss (blue line) consistently outperforms the one trained with MSE loss (orange line), achieving lower average MAE across all six LTF benchmarks. The inferior performance of the MSE-trained model can be attributed to rapid error accumulation during auto-regressive inference.

**Impact of Prediction Head Type.** Finally, we evaluated three prediction head designs (**M**ulti-**R**esolution(**MR**), **S**ingle-**R**esolution(**SR**), **M**ulti-**H**ead(**MH**)). As seen on the right, the **M**ulti-**R**esolution(**MR**) method consistently achieves the lowest average MSE on LTF benchmarks across all prediction lengths, demonstrating its superior capability.

## D.5 ROUTING VISUALIZATION

We provide additional figures to illustrate the expert selection proportions for all evaluated datasets. Visualization of LTF benchmarks across all prediction horizons are demonstrated in Figure 9, 10, 11 and 12. Visualization of our proposed datasets are displayed in Figure 13, 14, 15 and 16. Instead of uniform usage, the **Sample-wise Top-k Router** consistently directs time series to a small subset of preferred experts, validating that they develop distinct specializations. This specialization exhibits two key properties: **Hierarchical Refinement**: Expert utilization sharpens through the layers. In line with our *backcast-forecast residual stacking* 3.2, as initial temporal patterns are modeled and removed, routing in deeper layers becomes more decisive, targeting experts specialized for the more subtle residual signals. This is evident in datasets like ETTh1, where routing concentration increases significantly from Layer 1 to Layer 2. **Dataset-Adaptive Specialization**: The roles of experts are context-dependent, adapting to the unique statistical properties of each dataset. For instance, the dominant experts for ETTh1 differ from those for ETTm1 and Weather, showcasing the flexibility in allocating computational resources. In summary, this dynamic, sample-level routing enables a powerful form of conditional computation. By dispatching time series to the most relevant sequence of specialized experts, PatchMoE efficiently models diverse temporal patterns, which is a key driver of its strong performance.

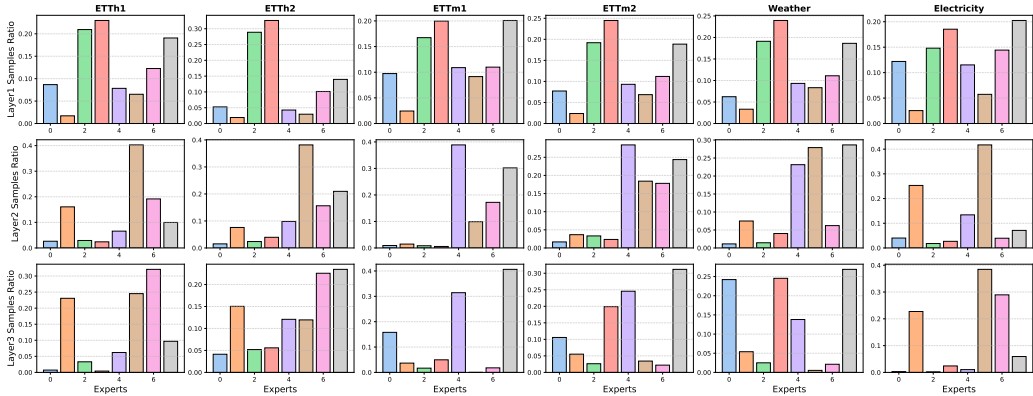

Figure 9: Visualization of the distribution of expert allocation of PatchMoE$_{\text{ultra}}$ on LTF benchmarks with seq-len 2880 and pred-len 96.

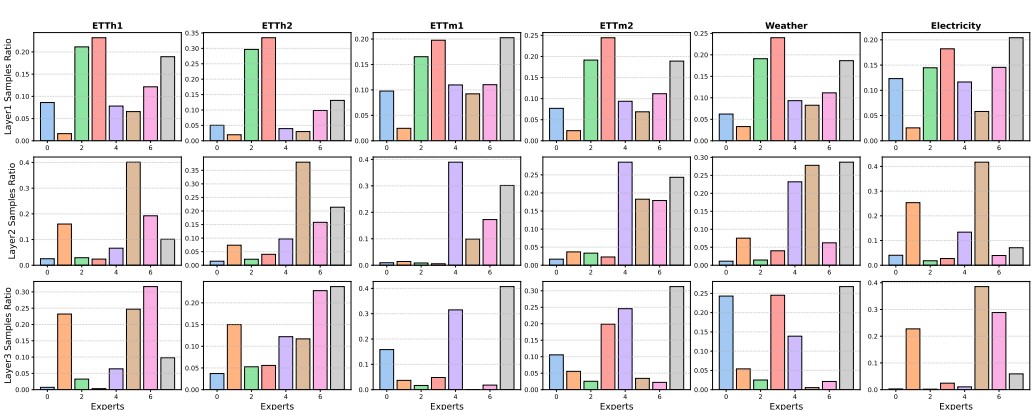

Figure 10: Visualization of the distribution of expert allocation of PatchMoE$_{ultra}$ on LTF benchmarks with seq-len 2880 and pred-len 192.

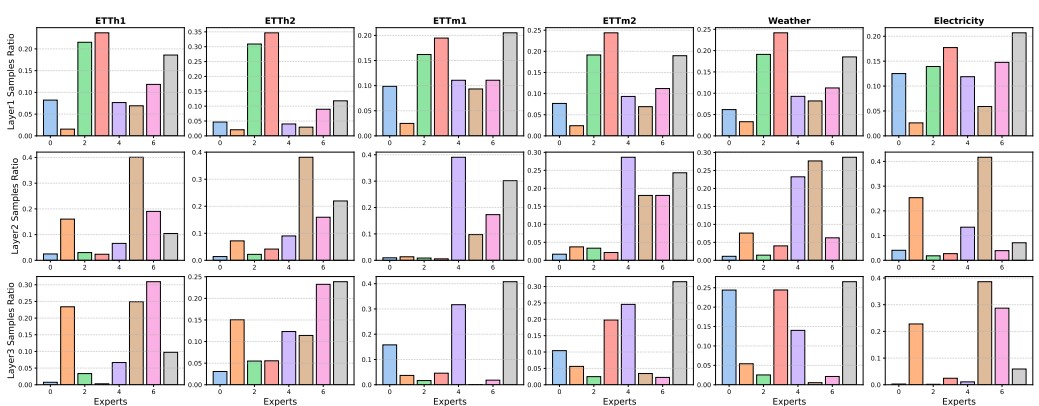

Figure 11: Visualization of the distribution of expert allocation of PatchMoE$_{ultra}$ on LTF benchmarks with seq-len 2880 and pred-len 336.

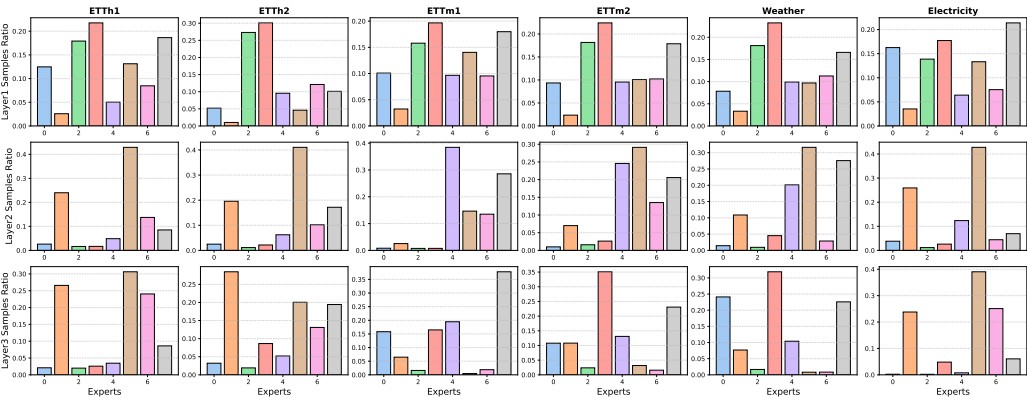

Figure 12: Visualization of the distribution of expert allocation of PatchMoE$_{ultra}$ on LTF benchmarks with seq-len 2880 and pred-len 720.

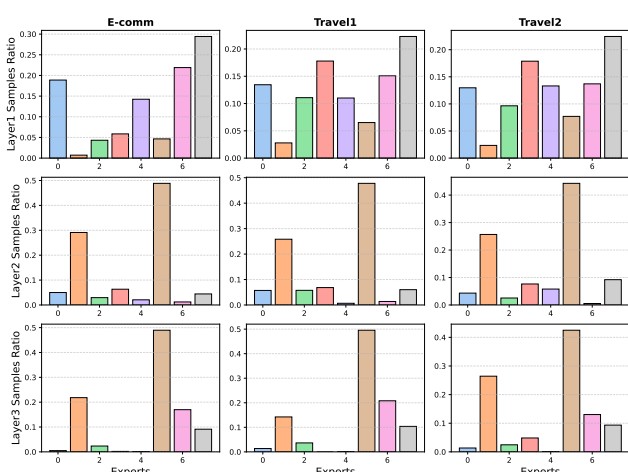

Figure 13: Visualization of the distribution of expert allocation of PatchMoE$_{\text{ultra}}$ on our proposed datasets with seq-len 2880 and pred-len 96.

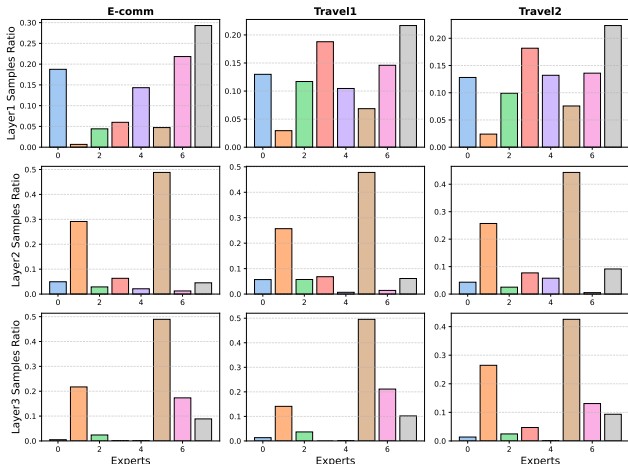

Figure 14: Visualization of the distribution of expert allocation of PatchMoE$_{\text{ultra}}$ on our proposed datasets with seq-len 2880 and pred-len 192.

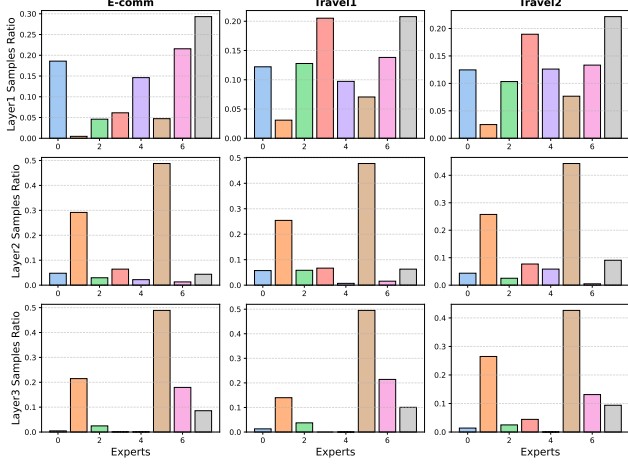

Figure 15: Visualization of the distribution of expert allocation of PatchMoE$_{\text{ultra}}$ on our proposed datasets with seq-len 2880 and pred-len 336.

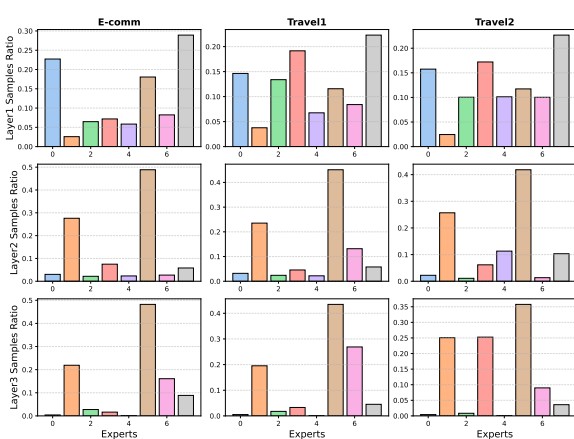

Figure 16: Visualization of the distribution of expert allocation of PatchMoE$_{ultra}$ on our proposed datasets with seq-len 2880 and pred-len 720.

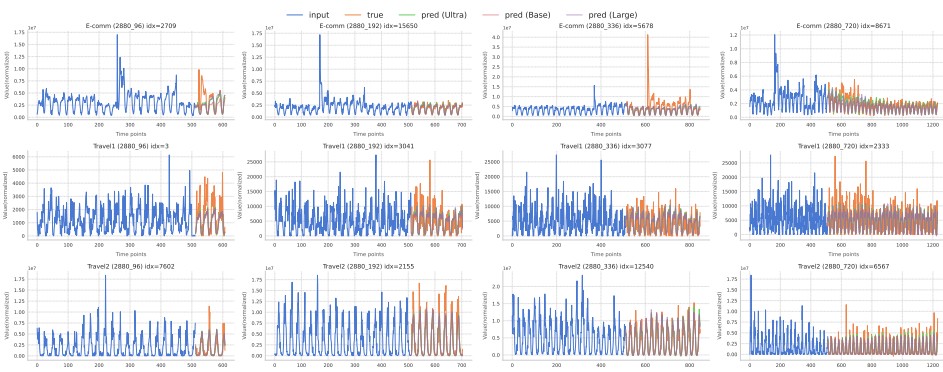

Figure 17: Showcases of zero-shot predictions from PatchMoE on our proposed datasets.

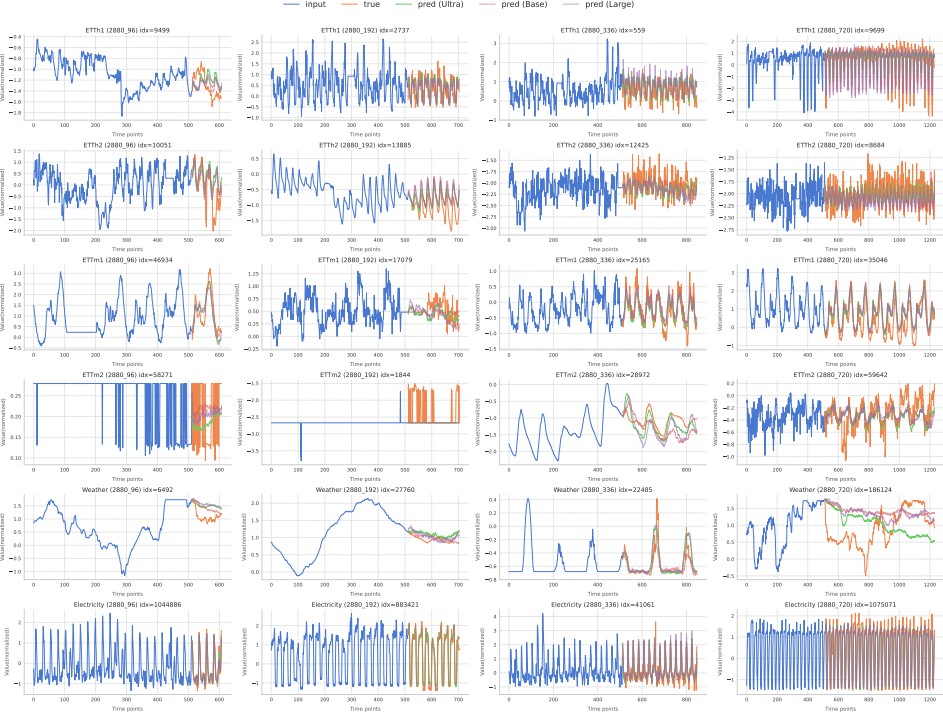

Figure 18: Showcases of zero-shot predictions from PatchMoE on LTF benchmarks.

### D.6  Show Cases

To provide a qualitative assessment of our model's forecasting capabilities, we present visualization showcases in Figure 17 and 18. These figures display prediction results on both our proposed datasets (E-comm, Travel1, Travel2) and widely-used long-term forecasting benchmarks (ETTh1, ETTh2, ETTm1, ETTm2, Weather, Electricity). We visualize the complete set of prediction configurations: each column corresponds to a distinct prediction horizon $T \in \{96, 192, 336, 720\}$, while the input look-back window is consistently fixed at $L = 2880$. The plots overlay the predictions from our three model variants—Base, Large, and Ultra—which represent different model capacities. These visualizations affirm our model's effectiveness in handling diverse time series data for both short-term and challenging long-term forecasting tasks, underscoring the robustness and scalability of our architecture across different parameterization scales.

## E  The Use of Large Language Models

To be clear, the core methodology and all technical components of this work were developed strictly **without** the use of Large Language Models (LLMs). LLMs were solely used to assist with improving the language, clarity, and readability of the paper. No other contributions are made by LLMs.

