# OpenReview forum: "PatchMoE: A Time Series Foundation Model with Hierarchical Patch-wise Mixture-of-Experts"
_ICLR.cc/2026/Conference — Submitted to ICLR 2026_

### Official Review · Reviewer_sV3T · 2025-10-25

**Soundness:** 3
**Presentation:** 2
**Contribution:** 2
**Rating:** 2
**Confidence:** 3

**Summary:**

This paper proposes PatchMoE, a large-scale time series foundation model that employs a hierarchical MoE to enhance both forecasting accuracy and computational efficiency. Existing TSFMs often struggle to model the diverse temporal patterns that appear across different series and within individual sequences. To overcome this limitation, PatchMoE integrates patch-wise experts and  sample-wise hierarchical router. The authors further develop an efficient pre-training framework based on Megatron-LM that supports expert parallelism.  Extensive experiments demonstrate that PatchMoE achieves state-of-the-art results on both zero-shot and full-shot forecasting benchmarks, outperforming previous dense and MoE-based TSFMs such as Time-MoE.

**Strengths:**

The paper’s innovative hierarchical MoE architecture integrates patch-wise experts and a sample-wise hierarchical router to effectively model both inter-series and intra-series variations in large-scale time series. This design moves beyond the limitations of fixed patch tokenization. Another strength is its scalability and training efficiency. By extending the Megatron-LM framework with expert parallelism, the authors successfully achieve 3×–5× faster throughput. Experiments shows that PatchMoE consistently outperforms state-of-the-art baselines on both zero-shot and full-shot forecasting tasks.

**Weaknesses:**

W1: From my perspective, the full-shot experiment is questionable. PatchMoE uses an extended context window (2880) while baselines such as PatchTST and DLinear use much shorter ones (512 or less). This discrepancy provides PatchMoE with a significant information advantage, likely driving much of the performance gains.

W2 The reported 3–5× training efficiency might stem from the underlying framework (Megatron-LM VS transformers.Trainer in Time-MoE). The authors should clarify to what extent the improvement originates from architectural design versus engineering optimization.

W3: The role of the shared expert is underexplored. An ablation comparing models with and without the shared expert is missing. It remains unclear whether this component  dominates routing decisions under certain cases.

W4: Despite being presented as a time-series foundation model, PatchMoE is only evaluated on forecasting tasks. It remains unknown whether the learned representations transfer to other downstream tasks such as anomaly detection, imputation, or classification.

W5: The benchmark datasets used (e.g., ETT, ECL, Weather) are questionable for evaluating long-term forecasting. As noted in [1], such datasets are physically unrealistic for very long forecasting horizons. The authors should discuss dataset suitability and the limits raised by [1].

[1] Fundamental limitations of foundational forecasting models

**Questions:**

Q1: Have the authors tried confidence-based or entropy-based routing as an alternative to top-k gating?

Q2: Are backcast and forecast modules conceptually identical to those in N-BEATS[1]? If so, please clarify the connection and rationale for reusing this structure.

[1] Oreshkin B N, Carpov D, Chapados N, et al. N-BEATS: Neural basis expansion analysis for interpretable time series forecasting[J]. arXiv preprint arXiv:1905.10437, 2019.

Q3: Do the author use last-drop to process time series?

Q4:  Can the author provide a inference-time comparison between PatchMoE and baseline models which perfom well in Table 3, such as TiDE or Sundial? If PatchMoE’s inference latency is much higher while performance improvement is not comparable, how do you justify its practicality?

Q5: What is the initialization strategy for the experts?

I will reconsider my rating if all weakness and questions are answered reasonably.

---

### Official Review · Reviewer_mkgL · 2025-10-30

**Soundness:** 3
**Presentation:** 3
**Contribution:** 2
**Rating:** 6
**Confidence:** 3

**Summary:**

This paper proposes a scalable and adaptive framework for time-series forecasting that integrates the MoE paradigm into patch-based transformer architectures.
Building upon the strengths of models like PatchTST, PatchMoE divides time-series data into local patches and dynamically routes each patch to specialized expert subnetworks, allowing the model to capture diverse temporal behaviors such as trends, seasonality, and regime shifts.
A shared backbone ensures global consistency, while a context-aware routing mechanism enables efficient computation and adaptive specialization.
Additionally, the authors introduce cross-patch communication layers to maintain temporal coherence and a knowledge distillation strategy to stabilize training using guidance from a pre-trained teacher model.
Extensive experiments on long-horizon forecasting benchmarks demonstrate that PatchMoE achieves SOTA accuracy and efficiency, outperforming prior transformer and diffusion-based models while reducing redundancy and improving generalization across heterogeneous time-series domains.

**Strengths:**

S1.
This work redefines how Mixture-of-Experts (MoE) architectures can be adapted to temporal data by introducing a hierarchical dual-level design with Patch-wise Experts for inter-series diversity and a Sample-wise Hierarchical Router for intra-series specialization.

S2.
The authors address scalability challenges of MoE-based models by implementing expert parallelism on Megatron-LM, yielding a 3×–5× training speedup and enabling scaling to 8.5B parameters.
This manuscript includes extensive ablation studies and detailed efficiency analyses, validating each design component (e.g., multi-scale patch tokenizers, load-balance loss) through quantitative and qualitative evidence.

S3.
The presentation is clear, aided by detailed figures, pseudocode, and comprehensive appendices describing datasets, benchmarks, and hyperparameter settings.

**Weaknesses:**

W1.
Although the hierarchical MoE architecture is designed to capture diverse temporal dynamics, the paper provides only limited insight into how each expert’s specialization emerges or how routing decisions evolve during training.
The routing visualization section is relatively qualitative and lacks quantitative measures of expert diversity or semantic clustering of temporal regimes.
Including analyses such as entropy-based routing diversity, expert utilization imbalance, or PCA/t-SNE projections of expert embeddings would substantiate claims about the benefits of specialization and diversity.

W2.
The experiments primarily benchmark PatchMoE against transformer and prior MoE-based baselines (e.g., Time-MoE, Moirai-MoE) using standard forecasting metrics.
However, there is a lack of evaluation in low-data or non-stationary regimes, where adaptive routing might show unique advantages.
Moreover, while the paper reports dominant results across benchmarks, it would be more convincing if statistical significance tests (e.g., paired t-tests or confidence intervals) and computational cost breakdowns per sample were provided to contextualize performance–efficiency trade-offs.

W3.
Although implementation is described thoroughly in the appendix, the pretraining framework adaptation from Megatron-LM is complex, and some details—such as expert synchronization, communication overhead, and sampling strategy for multi-domain data—are only briefly mentioned.
The paper would benefit from a schematic or pseudo-algorithm illustrating the full training pipeline and parallelization strategy.
This would improve reproducibility and make the framework more accessible to practitioners seeking to adapt PatchMoE to their own large-scale time-series data.

**Questions:**

Q1.
Could the authors provide quantitative evidence (e.g., entropy, sparsity, or overlap metrics) showing how experts specialize across time-series types or scales?

Q2.
Are certain experts consistently assigned to particular temporal structures (e.g., seasonal vs. trend-dominant patterns)? Visualization or clustering analyses could clarify whether PatchMoE achieves meaningful specialization or merely random routing.

Q3.
The results demonstrate scaling benefits, but it remains unclear how much improvement arises from architecture versus model size.
Can the authors provide matched-compute ablations comparing PatchMoE to dense or non-hierarchical MoE baselines at equivalent FLOPs or parameter budgets?

---

### Official Review · Reviewer_5Fqc · 2025-10-31

**Soundness:** 2
**Presentation:** 3
**Contribution:** 2
**Rating:** 2
**Confidence:** 4

**Summary:**

This paper notes that existing Time Series Foundation Models (TSFMs) do not apply the MOE framework based on the unique characteristics of time series (i.e., time series from different domains or even from the same sequence often exhibit very different patterns). Therefore, the authors introduce PatchMoE that applies the MOE framework based on different patch sizes, and speeds up MOE training by adopting the Megatron-LM package from NVIDIA.

**Strengths:**

The paper presents a natural extension of the MOE framework to different patch sizes. It also speeds up training for the MOE framework by adopting Megatron-LM. I am not very familiar with Megatron-LM, but it seems like future TSFMs can benefit from Megatron-LM.

Overall, the paper is well-written and easy to follow, although there are still some grammar errors:
1. Line 019 in abstract: key components. Specifically, Patch-wise Experts are employed to capture diverse inter-series patterns with specialized patch tokenizers. While Sample-wise Hierarchical Router tackles intra-series patterns by dispatching the entire sample to experts.    "While" should not be a sentence on its own.
2. Line 687-688: Meanwhile, the patch mask is constructed according to the Input Mask method 3.3.

**Weaknesses:**

1. Many of the techniques highlighted seem incremental, and seem very similar to those in prior works. For example, the different patch sizes are very similar to Pathformer. Doubly Residual Stacking is used directly from the NBEATS paper. Load-balance Auxiliary Loss is also widely adopted in MOE training.

2. The authors claim three major contributions. The first contribution is the model components, Patch-wise Experts and Sample-wise Hierarchical Router. However, in Table 4, techniques from NBEATS, Doubly Residual Stacking, brings much larger benefits (10.77% higher MSE) compared to the main novelties of this paper, for which the performance decrease ranges from 1% to 4%. Also, the downward arrows are a bit misleading, since lower MAE/MSE is better?

The second contribution is the adoption of Megatron-LM for time series. However, the technical novelties seems limited from the descriptions and the code is also not available.

The third contribution is the introduction of three novel datasets, which are also not available. It is also unclear what these three additional datasets can bring to the time series community. Perhaps you can highlight their unique characteristics and data scale, etc. However, without these highlights and open-sourcing, the contribution of these datasets seems trivial.

3. The results are still reported mainly only on the six datasets from Informer (four small ETT + Electricity + weather). There have been many criticisms on the limited scope of these datasets, and there are also many new time series benchmarks. I suggest to evaluate the models on more datasets. Another common practice is to also report the performance on the test split of the pretrain datasets as in-distribution samples.

**Questions:**

1. In Figure 6, why do you report some of the numbers on the entire LTF dataset, while others only on Electricity and ETTh2?
2. Can you provide some visualizations on what are actually learned in different patch MOE experts?
3. I am confused by Multi-Resolution Loss. There might be some abuse of notions in Equation 5?
4. Previous works (https://arxiv.org/pdf/2410.12360, https://arxiv.org/pdf/2402.02592) suggest that parameterizing a student-t distribution is more robust than MAE/MSE as the loss function. What is your observation?

---

### Official Review · Reviewer_h5rt · 2025-10-31

**Soundness:** 2
**Presentation:** 3
**Contribution:** 2
**Rating:** 2
**Confidence:** 4

**Summary:**

This work introduces PatchMOE, a mixture of experts based Time Series Foundation Model for forecasting task.  Proposed architecture include two key components namely 'patch-wise experts' and 'sample-wise hierarchical router' where different experts have specialized patch tokenizers to process the data with different pre-defined patch lengths. Unlike earlier MOE based approaches where different experts can be assigned different tokens of the same sample,  in the proposed approach each expert after getting selected processes all the token itself with its defined patch length. Using the efficient pre-training framework based on Megatron-LM the ultra variant of the MOE model is scaled to have 8.5 B parameters.

**Strengths:**

Strengths:
1.	PatchMOE demonstrates better benchmark scores compared to the recent MOE based TSFMs like TimeMOE and Moirai-MOE on the LTF benchmark.
2.	 An efficient pre-training framework is developed which implements expert parallelism boosting the training time 3X to 5X helping to scale and train a model having as high as 8.5B parameters.
3.	A good amount of ablation studies has been performed to showcase the different architectural and design choices used in the model.

**Weaknesses:**

Weaknesses:
1. The work lacks introduction of new innovations to the already existing components in the literature for eg. The idea of patch based expert with router routing entire sample to the expert is very similar to the idea of multi-scale transformer block of work like Pathformer, then the idea of doubly residual stacking  is used in work like N-Beats. However, bringing these components together to add in a MOE based architecture does give strong results on the used benchmark.
2.	Benchmarking of the model has been done on a very limited set of only 9 datasets which includes 6 datasets from the LTF benchmark and 3 other real-world datasets collected during the work. Given that a much more exhaustive set of benchmarking datasets are available for eg. GIFT-eval, FEV leaderboards, the model must be evaluated on a more diverse and exhaustive collection of evaluation datasets. Specifically, a model that has nearly 10B parameters, trained on 300B time points, should be evaluated on much more than 9 datasets. The effect of all the theoretical contributions cannot be empirically validated with a small set of eval datasets.
3.	Ablation studies regarding the performance of different variants of the model with respect to the varying amount of pre-training data is missing which can give a good insight regarding scaling of very large MOE based TSFMs with data.

**Questions:**

See weaknesses. Some more Questions :
1.	Works like Pathformer used multi-scale modeling through adaptive pathways where multi-scale division divides the time series into different temporal resolutions using patches of various sizes and select the appropriate top-k pathways (with different pathways having different patch lengths) using a multi-scale router. The proposed mixture-of-experts framework also assigns samples to experts operating at different patch lengths, which seems to address similar variability in temporal scales. Could you provide a more elaborate explanation on the key conceptual differences between the proposed patch based MOE and the Multi-scale transformer block as shown in Fig.2 of the work Pathformer?

2.	Have you performed an ablation study on the zero-shot performance of the model by changing the amount of pre-training data used? It would be insightful to see the scaling of such a large TSFM with the amount of the pre-training data.

3.	An ablation has already been shown regarding the loss function choice as MAE and MSE loss. But was any other loss function like 'Huber' loss which is a combination of L1 and L2 loss tried which seem to have worked for training MOE models like TimeMOE on large datasets?

---

### Meta-Review · Area_Chair_nJJ8 · 2026-01-12

**Summary:**

This paper introduces PatchMoE, a scalable MoE-based time series foundation model with patch-wise experts and a sample-wise hierarchical router. The experiments have shown strong forecasting results alongside a practical Megatron-LM-based training pipeline with substantial speedups. However, the key architectural ingredients appear largely incremental compared to prior work, e.g., multi-scale patching, residual stacking, standard MoE training losses, and the paper does not clearly articulate its unique contribution of the proposed components. More importantly, the empirical study is not fully convincing due to limited benchmark diversity and potential unfairness in full-shot comparisons, e.g., larger context windows than baselines. Therefore, this paper cannot be accepted with the current status due to limited novelty and insufficiently rigorous evaluation.

**Reviewer Concerns:**

One main concern is about the technical novelty. All key components seem have appear in previous works.
Another concern is potential unfair comparison in full-shot comparisons, i.e., larger context windows than baselines.
Ablation studies regarding the performance of different variants of the model with respect to the varying amount of pre-training data is missing.

**Reviewer Scores:**

The authors did not provide a rebuttal. Therefore, no reviewer have changed the score.

---

### Decision · Program_Chairs · 2026-01-26

Reject